# The Hippo pathway effector YAP is an essential regulator of ductal progenitor patterning in the mouse submandibular gland

Aleksander D Szymaniak[1], Rongjuan Mi[1,2], Shannon E McCarthy[1], Adam C Gower[3], Taylor L Reynolds[4], Michael Mingueneau[4], Maria Kukuruzinska[2], Xaralabos Varelas[1]*

[1]Department of Biochemistry, Boston University School of Medicine, Boston, United States; [2]Department of Molecular and Cell Biology, Boston University School of Dental Medicine, Boston, United States; [3]Clinical and Translational Science Institute, Boston University, Boston, United States; [4]Immunology Research, Biogen, Cambridge, United States

**Abstract** Salivary glands, such as submandibular glands (SMGs), are composed of branched epithelial ductal networks that terminate in acini that together produce, transport and secrete saliva. Here, we show that the transcriptional regulator Yap, a key effector of the Hippo pathway, is required for the proper patterning and morphogenesis of SMG epithelium. Epithelial deletion of *Yap* in developing SMGs results in the loss of ductal structures, arising from reduced expression of the EGF family member Epiregulin, which we show is required for the expansion of Krt5/Krt14-positive ductal progenitors. We further show that epithelial deletion of the *Lats1* and *Lats2* genes, which encode kinases that restrict nuclear Yap localization, results in morphogenesis defects accompanied by an expansion of Krt5/Krt14-positive cells. Collectively, our data indicate that Yap-induced Epiregulin signaling promotes the identity of SMG ductal progenitors and that removal of nuclear Yap by Lats1/2-mediated signaling is critical for proper ductal maturation.

*For correspondence: xvarelas@bu.edu

## Introduction

The mammalian epithelial branching program is a highly dynamic and organized process that leads to the formation of branched networks of tubule structures that terminate in acini with specialized functions. Understanding how epithelial progenitor cells pattern into ducts and acini, and how this is coordinated with ongoing tissue morphogenesis, is one of the central questions in epithelial development. The submandibular gland (SMG) offers a model to study the molecular mechanisms directing epithelial branching morphogenesis and patterning, with distinct synchronized processes of cell proliferation, clefting, differentiation, migration and apoptosis occurring rapidly during embryogenesis (*Hauser and Hoffman, 2015*; *Mattingly et al., 2015*). The developing SMG epithelium communicates with neighboring mesenchymal, neuronal and endothelial cells to direct reiterative rounds of bud and duct formation that mature into epithelial domains that mediate the production, transportation, and secretion of saliva (*Knosp et al., 2015*; *Knox et al., 2010*; *Lombaert et al., 2013*; *Patel et al., 2011*; *Steinberg et al., 2005*; *Wells et al., 2014*). Following initial bud formation, it is thought that specification of distinct multipotent progenitor populations give rise to the specialized cell populations that compose the acinar and ductal domains. For example, multi-potent populations of Cytokeratin-5 (Krt5, K5)- and Cytokeratin-14 (Krt14, K14)-positive progenitors are thought to

**eLife digest** Our mouths are continually bathed by saliva – a thick, clear liquid that helps us to swallow and digest our food and protects us against infections. Saliva is produced by and released from salivary glands, which are organs that contain a branched network of tubes. Salivary glands can only properly develop if immature cells known as stem cells, which give rise to the mature cells in the organ, are controlled. Despite their importance for development of salivary glands, little has been known about the signals that control these stem cells.

Szymaniak et al. have now discovered new regulators of the salivary gland stem cells in mice, including essential roles in the regulation of these cells by a protein known as Yap. The Yap protein is controlled by a set of proteins that together are known as the Hippo pathway. Szymaniak et al. found that when the gene for Yap was deleted in mice very few stem cells were made, and the transport tubes of the salivary tubes failed to develop. Conversely, when the Hippo pathway was disrupted in mice there were too many stem cells because they could not properly develop into the mature cells, leading to incorrect transport tube development..

These results indicate that Yap is essential for controlling the stem cells of the salivary glands, and offer important insight into the signals that control how the salivary glands develop. The next step will be to investigate whether the Hippo pathway or Yap are affected in diseases of the salivary gland, which often show incorrect numbers of stem cells.

govern the formation of epithelial cells that give rise to the mature ductal structures (*Knox et al., 2010*; *Lombaert et al., 2013*). Although numerous molecular studies have focused on understanding the biology of SMG progenitors, much remains unclear about the intrinsic signals that define their identity and/or control their differentiation.

Recent studies have provided evidence that the transcriptional regulator Yap plays essential roles in stem cell biology and that these roles are essential for the development of branching organs, such as the kidney, lung, pancreas, and mammary gland (*Varelas, 2014*). Ectopic expression of *Yap* has been shown to drive the expansion of progenitor populations in several tissues, while conditional deletion of *Yap* in organ-specific stem cells can lead to the inhibition of stem cell specification or the induction of premature differentiation (*Mahoney et al., 2014*; *Panciera et al., 2016*; *Zhao et al., 2014*). In particular, the dynamics of Yap localization is implicated in controlling the balance of specification, self-renewal, and differentiation of various stem cell populations. Yap localization is controlled by a multitude of signals that include those mediated by the Hippo pathway (*Meng et al., 2016*). The Hippo pathway is comprised of a series of kinase-mediated signaling events that result in the phosphorylation and activation of the Lats1 and Lats2 kinases (herein together referred to as Lats1/2). Activated Lats1/2 redundantly direct the phosphorylation of Yap on conserved serine residues, the best characterized of which is S112 in mouse Yap (S127 in human Yap), which restricts the nuclear accumulation and transcriptional activity of Yap. Loss of Lats1/2-mediated regulation of Yap activity leads to defective organ patterning and function (*Heallen et al., 2011*, *2013*; *Reginensi et al., 2016*; *Yi et al., 2016*), highlighting the importance of Hippo pathway signaling in development.

Here, we used genetic approaches to examine the roles of Hippo-Yap signaling in SMG epithelial development. We found that embryonic deletion of Yap in developing SMGs resulted in severe morphogenesis defects, which notably did not arise from aberrant cell proliferation or apoptosis, but rather from defects in progenitor patterning. We show that Yap is required for the specification of Krt5/Krt14-positive ductal progenitor cells, and that Yap does so, in part, by controlling the expression of the epidermal growth factor family member Epiregulin (Ereg). Treatment of ex vivo cultured SMGs with Ereg was sufficient to expand Krt5/Krt14-positive cells and rescue cell fate specification defects observed in *Yap*-deleted SMGs. Conversely, we found that deletion of the Hippo kinases *Lats1/2* resulted in massive expansion of Krt5/Krt14-positive ductal cells in developing SMG epithelium, and that this phenotype could be blunted by EGFR (ErbB-1) inhibition. These findings demonstrate that Yap is a critical regulator of ductal progenitor cell identity in SMG epithelium and that proper control of Yap localization by Lats1/2 is essential for the maturation of SMG ducts. Our study

therefore identifies novel essential effectors of SMG development and provides important insight into early patterning events that are coupled with branching morphogenesis.

## Results

### Nuclear Yap marks distinct populations of developing SMG ductal epithelial cells

To gain insight into the role(s) of Yap during SMG development we examined the levels and localization of Yap in early branching SMGs using immunofluorescence (IF) microscopy. We closely monitored the distribution of Yap with respect to known markers of early patterning, including Krt14, which labels multipotent progenitors that can give rise to ductal epithelial cells (*Knox et al., 2010*; *Nedvetsky et al., 2014*) (illustrated in *Figure 1A*). We found that Yap was prominently expressed in E13.5 SMG epithelium, and distinct Yap localization differences were apparent in various cell populations of the branching gland. A cell layer at the peripheral edge of each end-bud showed some cells with nuclear Yap localization, whereas all cells immediately adjacent and extending away from the edge of the bud showed cytoplasmic Yap localization (*Figure 1B–C*). Yap also showed very prominent nuclear localization in cells transitioning proximally towards the newly developing ductal regions (*Figure 1B–C*), overlapping precisely with Krt14-positive ductal progenitors (*Figure 1D*).

As the ductal epithelium of the SMG starts to mature, cells begin stratifying into luminal and basal layers. Krt5-positive basal-positioned cells possess stem cell activity and are believed to play important roles in adult SMG injury repair (*Knox et al., 2013*) (illustrated in *Figure 1E*). The large majority of maturing ductal epithelial cells in E15.5 and in differentiated E18.5 SMGs exhibited cytoplasmic Yap localization, particularly cells positioned at the luminal layer (*Figure 1F–G*). Cytoplasmic Yap localization correlated with increased Serine-112 phosphorylation of Yap (pS112-Yap) (*Figure 1F*), which is a site phosphorylated by the Hippo pathway kinases Lats1/2 and promotes cytoplasmic Yap localization (*Dong et al., 2007*). While few in number, some ductal epithelial cells in E15.5 and E18.5 SMGs exhibited prominent nuclear Yap localization, and these cells were generally positioned in the basal layer of Krt5-positive epithelial cells. However, not all Krt5-positive cells showed nuclear Yap localization, suggesting that nuclear Yap marks a distinct sub-population of cells with this marker or that we captured a snapshot of dynamic Yap localization occurring in these cells (*Figure 1F–G*).

### Epithelial deletion of Yap results in severe branching defects and impaired ductal domain specification in developing SMGs

To assess the importance of Yap in SMG development, we sought to conditionally delete *Yap* in the epithelium of embryonic SMGs. Cre recombinase driven by the promoter of the *Shh* (sonic hedgehog) gene (*Shh*-Cre) (*Harris et al., 2006*) has previously been used to target the SMG epithelium (*Knosp et al., 2015*), which prompted us to test the efficiency of this model for use in our studies. We started by crossing *Shh*-Cre mice with Rosa26-loxP-STOP-loxP-EYFP mice (*Srinivas et al., 2001*), allowing us to mark *Shh*-expressing cells with Enhanced Yellow Fluorescent Protein (EYFP). All *Shh*-Cre-positive/EYFP-positive SMGs that we examined showed robust EYFP signal marking the entire SMG epithelium (*Figure 2A*), indicating that the SMG epithelium originates from *Shh*-expressing cells and therefore this Cre model could be used to conditionally target loxP-flanked genes in the SMG epithelium. Accordingly, we found that crossing *Shh*-Cre mice with *Yap*-loxP/loxP mice led to the efficient deletion of *Yap* in the developing SMG epithelium (herein called *Yap*-cnull SMGs) (*Figure 2B*), and that this led to striking branching defects. E13.5 *Yap*-cnull SMGs lacked developed clefts and ducts (*Figure 2C–D*), and E15.5 *Yap*-cnull glands showed severely disorganized bud-like structures and complete absence of ductal trees (*Figure 2C–E*). Interestingly, these phenotypes did not appear to result from global increases in apoptosis (*Figure 2F*) or defects in overall epithelial proliferation (*Figure 2G*).

We hypothesized that the morphogenesis defects associated with *Yap* deletion may originate from compromised epithelial patterning, prompting us to examine the distribution of progenitor markers following *Yap* deletion. We first examined fixed E15.5 SMGs, which revealed an almost complete absence of Krt14-positive ductal progenitors (*Figure 3A*). To understand the dynamics of progenitor patterning, we isolated E13.5 SMGs from wild-type and *Yap*-cnull embryos and cultured them ex vivo for 24 hr. After 24 hr of explant culture, WT SMGs exhibited extensive branching, with

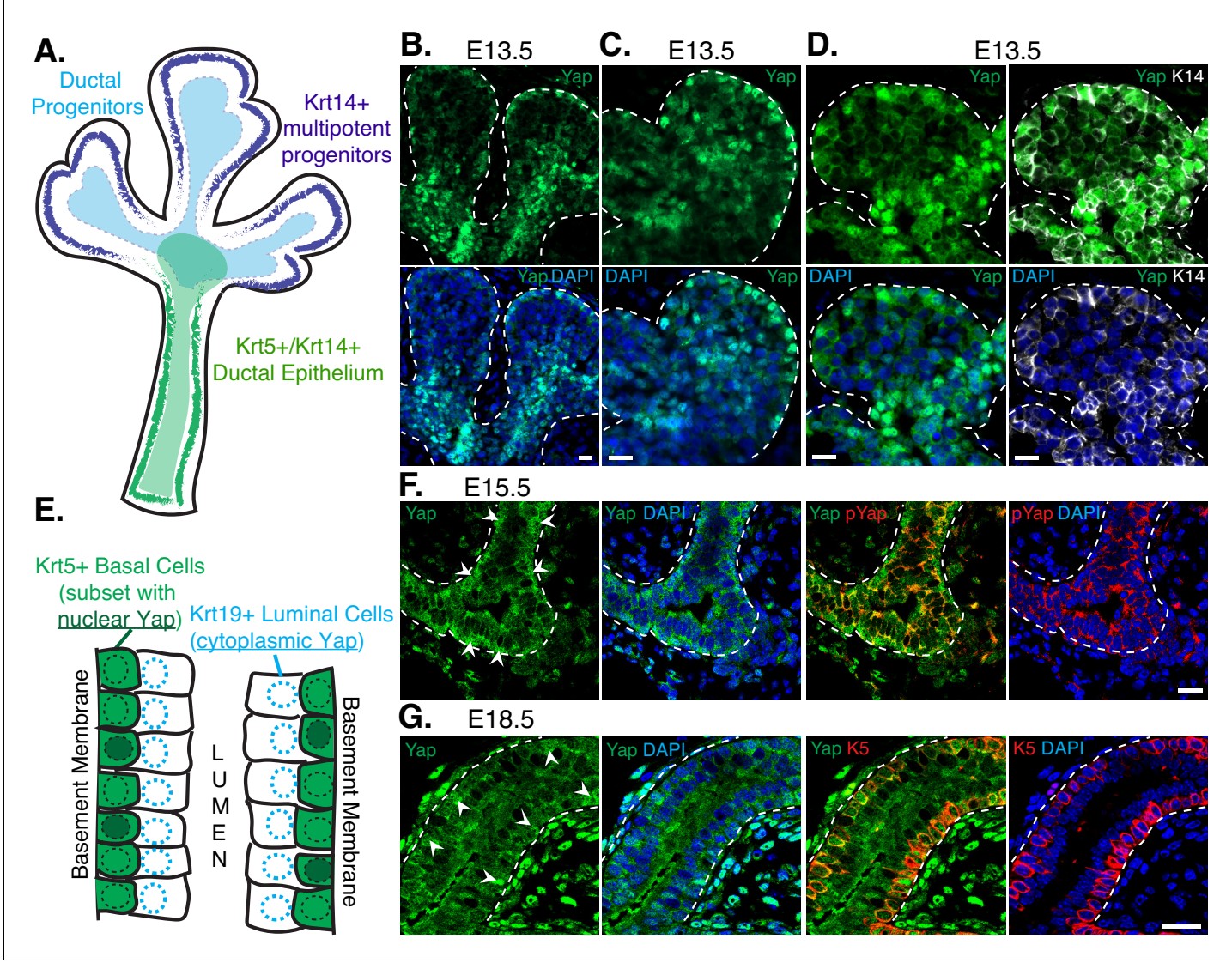

**Figure 1.** Nuclear Yap marks distinct populations of developing SMG ductal epithelial cells. (A) Illustration depicting the early developing SMG epithelium, with the positioning of relevant multipotent progenitor cells highlighted. (B,C) Two different magnified IF microscopy images of the bud-duct transition zone showing the localization of Yap (green) in E13.5 mouse SMG epithelium. (D) Images from IF microscopy analysis of Yap (green) together with Krt14 (K14, white) shows prominent nuclear Yap in Krt14-positive cells. (E) Illustration depicting the maturing stratified SMG ductal epithelium with relevant cell populations highlighted, and a description of our observed localization pattern for Yap. (F) Images from IF microscopy analysis of total Yap (green) and phospho-S112 Yap (red) in E15.5 mouse SMGs. In luminal cells, Yap phosphorylation levels are elevated and Yap is excluded from the nucleus, while many basal cells exhibit prominent nuclear Yap localization. (G) Images from IF microscopy analysis of E18.5 mouse SMG ducts for Yap (green) and Krt5 (K5, red) showing prominent nuclear Yap localization in a subset of Krt5-positive basal cells and cytoplasmic Yap in Krt5-negative luminal cells. White arrows highlight prominent nuclear Yap localization in the basal cells in (F) and (G). DAPI was used to mark the nuclei (blue) in all images, and for clarity the basal surface of the epithelium is outlined with a white dotted line. Scale bar = 20 µm. All images represent observations made from a minimum of three biological repeats.

expected robust Krt14 and Krt5 expression in developing ductal epithelial regions, and Krt19 expression marking maturing cells (*Figure 3B–E*). *Yap*-cnull SMGs failed to branch after 24 hr of culture and exhibited an almost complete absence of Krt5, Krt14, or Krt19-expressing cells, except for a small segment of the most proximal region (*Figure 3B–E*). The lack of these markers suggested that *Yap*-cnull SMGs fail to specify ductal progenitors and consequently ductal epithelium. Parasympathetic nerve innervation plays a crucial role in the growth and regulation of SMG ductal progenitors (*Knosp et al., 2015*; *Knox et al., 2010*; *Nedvetsky et al., 2014*), which prompted us to

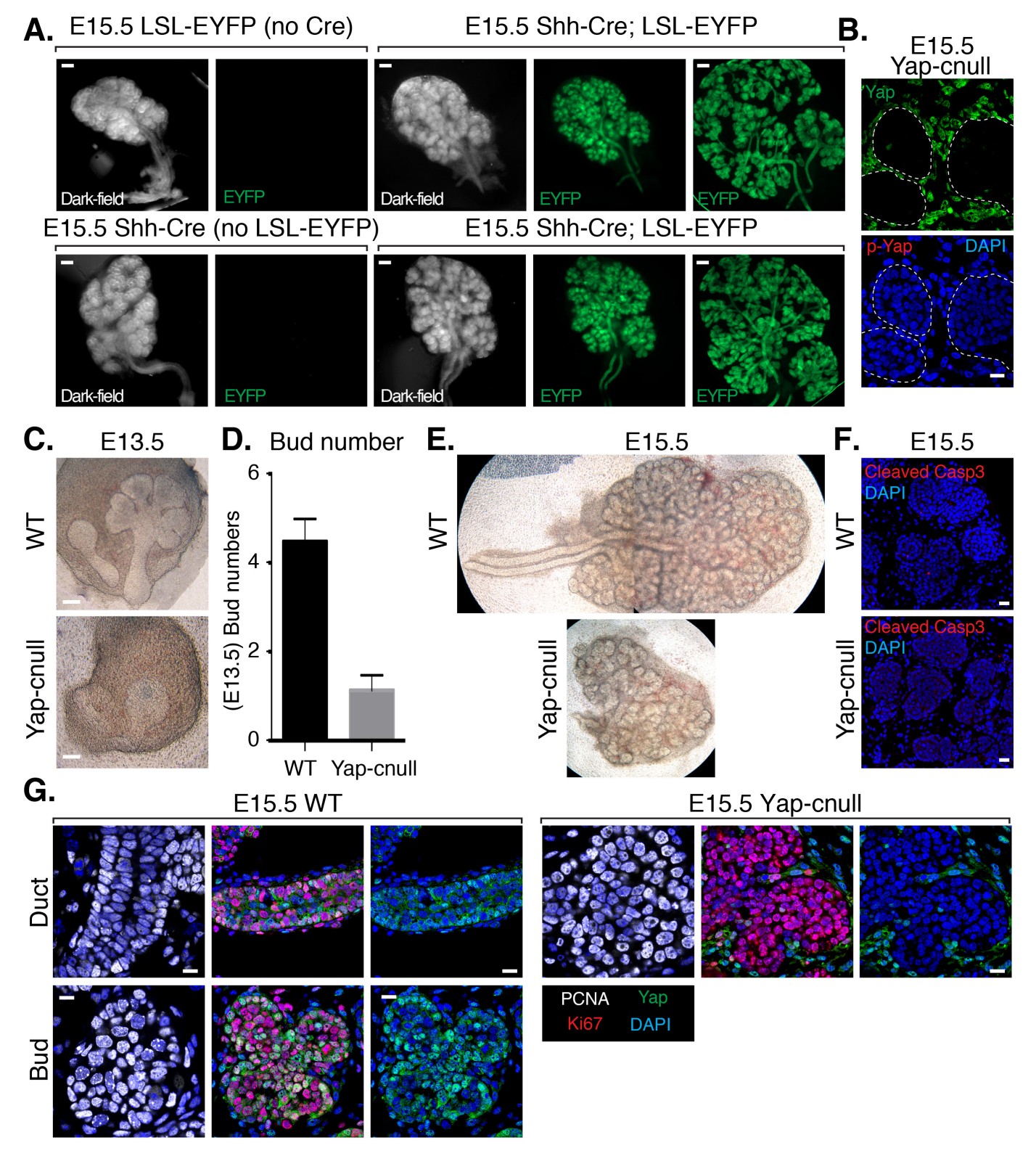

**Figure 2.** Deletion of *Yap* in developing SMG epithelium results in severe branching defects and defective ductal domain patterning. (**A**) Lineage tracing using *Shh*-Cre; ROSA26-lox-STOP-lox-EYFP reporter mouse. E15.5 mouse SMGs of the indicated genotypes were dissected and immediately imaged on an inverted microscope in dark-field for the left panels and for fluorescent EYFP signal in the right panels. In the furthest right panels, the SMGs were compressed under a coverslip to highlight the EYFP-positive epithelial branches. Scale = 200 μm (**B**) Images from IF microscopy imaging of

*Figure 2 continued on next page*

*Figure 2 continued*

E15.5 *Shh*-Cre-*Yap*-null (Yap-cnull) SMGs for total Yap (green) and phospho-S112 Yap (red) showing efficient deletion of Yap in the SMG epithelium (outlined by a dotted white line). (C) Phase-contrast images of E13.5 WT and *Yap*-cnull SMGs showing severe morphogenesis defects in *Yap*-cnull SMGs. Scale = 100 μm. (D) Quantitation of bud number from E13.5 WT and *Yap*-cnull SMGs (n = 23). (E) Phase-contrast images of E15.5 WT and *Yap*-cnull SMGs indicating a disorganized bud structure and lack of ductal trees in Yap-deficient SMGs. Note that the image from the WT SMG is stitched together from two images. (F) Images from IF microscopy analysis of Cleaved-caspase 3 in E15.5 WT and *Yap*-cnull SMGs showing no apparent defect in apoptosis. Note, that the Cleaved-caspase 3 antibody activity was validated in parallel slides containing positive cells. Scale = 20 μm. (G) Images from IF microscopy analysis of E15.5 WT and *Yap*-cnull SMGs for Yap (green), Ki-67 (red), or PCNA (white) showing no apparent proliferation defects in Yap-deleted epithelium. Scale = 10 μm. DAPI was used to mark the nuclei (blue). All images represent observations made from a minimum of three biological repeats.

examine the distribution of the parasympathetic nerve in *Yap*-cnull SMGs by staining for the nerve marker TuJ1. Despite being present in the *Yap*-cnull SMGs, parasympathetic nerve innervation was diminished and unorganized, suggesting that signaling crosstalk with the nerve may be compromised (*Figure 3F–G*). The structural organization of the actin cytoskeleton was also severely disrupted in *Yap*-cnull SMGs (*Figure 3G*), indicating defective polarization that is required for proper epithelial maturation. Interestingly, markers associated with the developing bud domains, such as Sox10, were abundant in *Yap*-cnull SMGs (*Figure 4A–C*), suggesting that the early development of the bud epithelium is unaffected following the loss of Yap. Taken together, our analyses indicated that the deletion of *Yap* leads to a loss of an early ductal progenitor population, resulting in branching morphogenesis defects.

## Global gene expression analysis identifies Epiregulin as an important signaling effector downstream of Yap that controls ductal progenitor specification

To gain insight into how Yap directs SMG development, we isolated RNA from three wild type and three *Yap*-cnull E15.5 SMGs (across three litters) and analyzed global gene expression using microarrays. Differential expression analysis using a stringent cutoff (FDRq <0.01, and fold change of two-fold or greater) revealed 105 genes that differed between the *Yap*-cnull and wild-type SMGs, one of which expectedly was *Yap* (*Figure 5—source data 1*). Hierarchical clustering of these genes showed that the expression profiles from replicate samples clearly clustered next to each other, with two major clusters of genes showing either reduced or increased expression in *Yap*-cnull SMGs (*Figure 5A*). Consistent with IF microscopy, *Krt5* and *Krt14* expression in ductal SMG progenitors was significantly reduced in the absence of Yap, which we further validated by quantitative real-time PCR (qPCR) (*Figure 5B*). Several canonical target genes of Yap, such as *Ctgf* and *Cyr61*, were among genes significantly reduced in *Yap*-cnull SMGs, validating that our data reflects Yap-driven gene expression.

Functional annotation clustering of the differentially expressed genes in *Yap*-cnull SMGs using DAVID Bioinformatics Resources (*Huang et al., 2009*) revealed enrichment of several interesting clusters of genes (*Figure 5C*). *Yap*-cnull SMGs were enriched in genes encoding secreted factors, suggesting that Yap plays an important role in altering the microenvironment of SMG epithelium. Enriched among the repressed genes were genes linked to Hippo signaling and those encoding factors with EGF-like domains. Additionally, genes encoding factors associated with the control of stem cell pluripotency were repressed in *Yap*-cnull SMGs. Conversely, genes linked to promoting cell differentiation were induced in *Yap*-cnull SMGs. Similar enrichment for genes encoding secreted factors or stem cell regulators was obtained when differentially expressed genes were examined by Gene Set Enrichment Analysis (*Figure 5—source data 2*) (*Subramanian et al., 2005*). This analysis revealed a significant negative correlation between genes repressed in *Yap*-cnull SMGs and genes involved in the negative regulation of cell differentiation (i.e. genes normally induced by Yap in a wild type setting positively correlate with genes that prevent cell differentiation) (*Figure 5D*). Thus, Yap-mediated transcription has a role in preventing cell differentiation and that genes regulated by Yap likely playing an important role in controlling the specification of multipotent ductal progenitors.

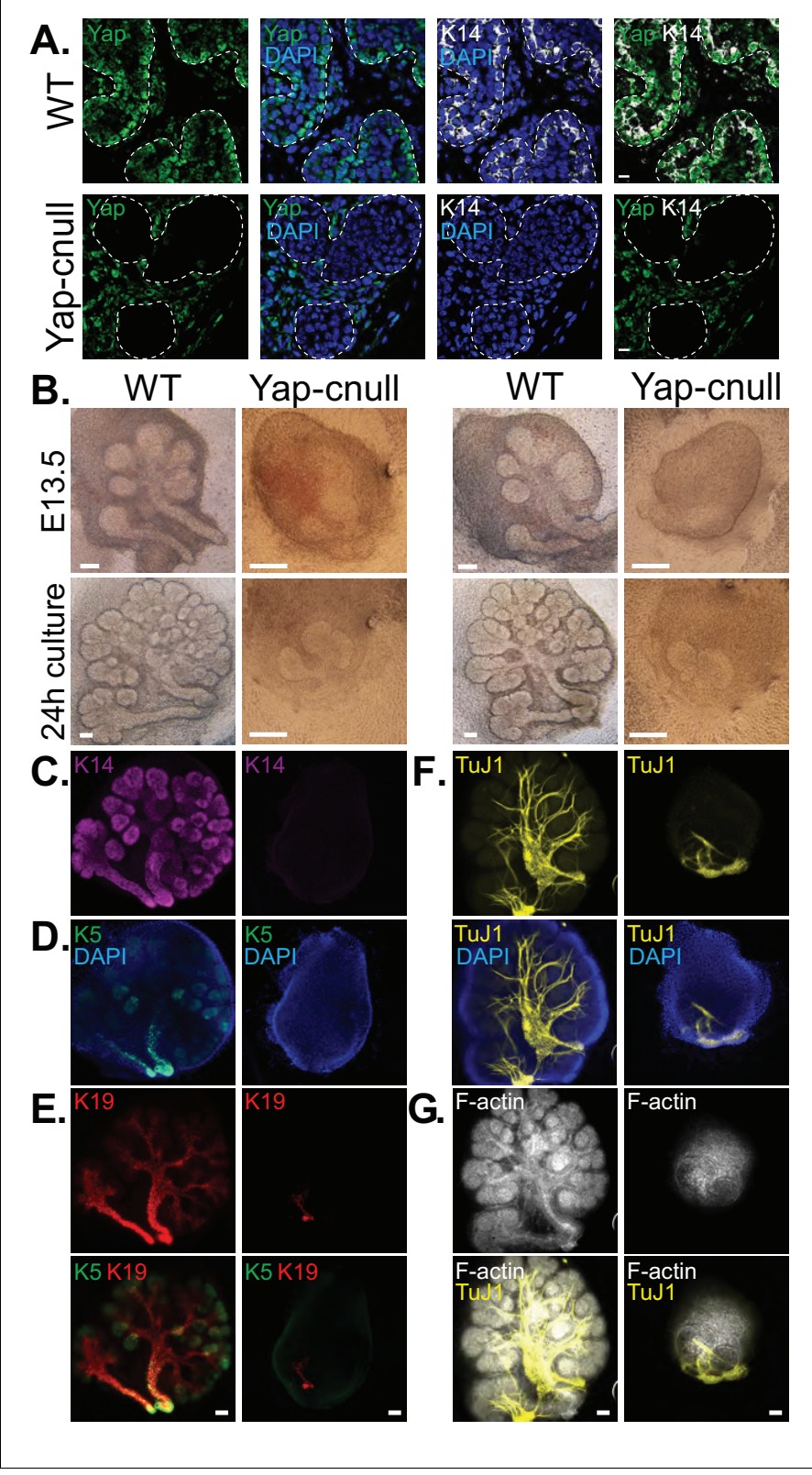

**Figure 3.** Yap is required for SMG ductal epithelial patterning. (**A**) IF analysis of Yap (green) and Krt14 (K14, white) in E15.5 WT and *Yap*-cnull SMGs. Scale = 10 μm. (**B–G**) E13.5 WT and *Yap*-cnull SMGs were dissected and cultured for 24 hr ex vivo and then examined by microscopy. (**B**) Phase-contrast images of E13.5 WT and *Yap*-cnull SMGs at the time of dissection and 24 hr after culture. The same SMGs (each column is one SMG) were analyzed

*Figure 3 continued on next page*

*Figure 3 continued*
by IF for (**C**) Krt14 (K14, magenta), (**D**) Krt5 (K5, green), (**E**) Krt19 (K19, red), (**F**) TuJ1 (yellow), and (**G**) F-actin (white, Phalloidin). DAPI was used to mark the nuclei (blue). Scale = 100 μm. All images represent observations made from a minimum of three biological repeats.

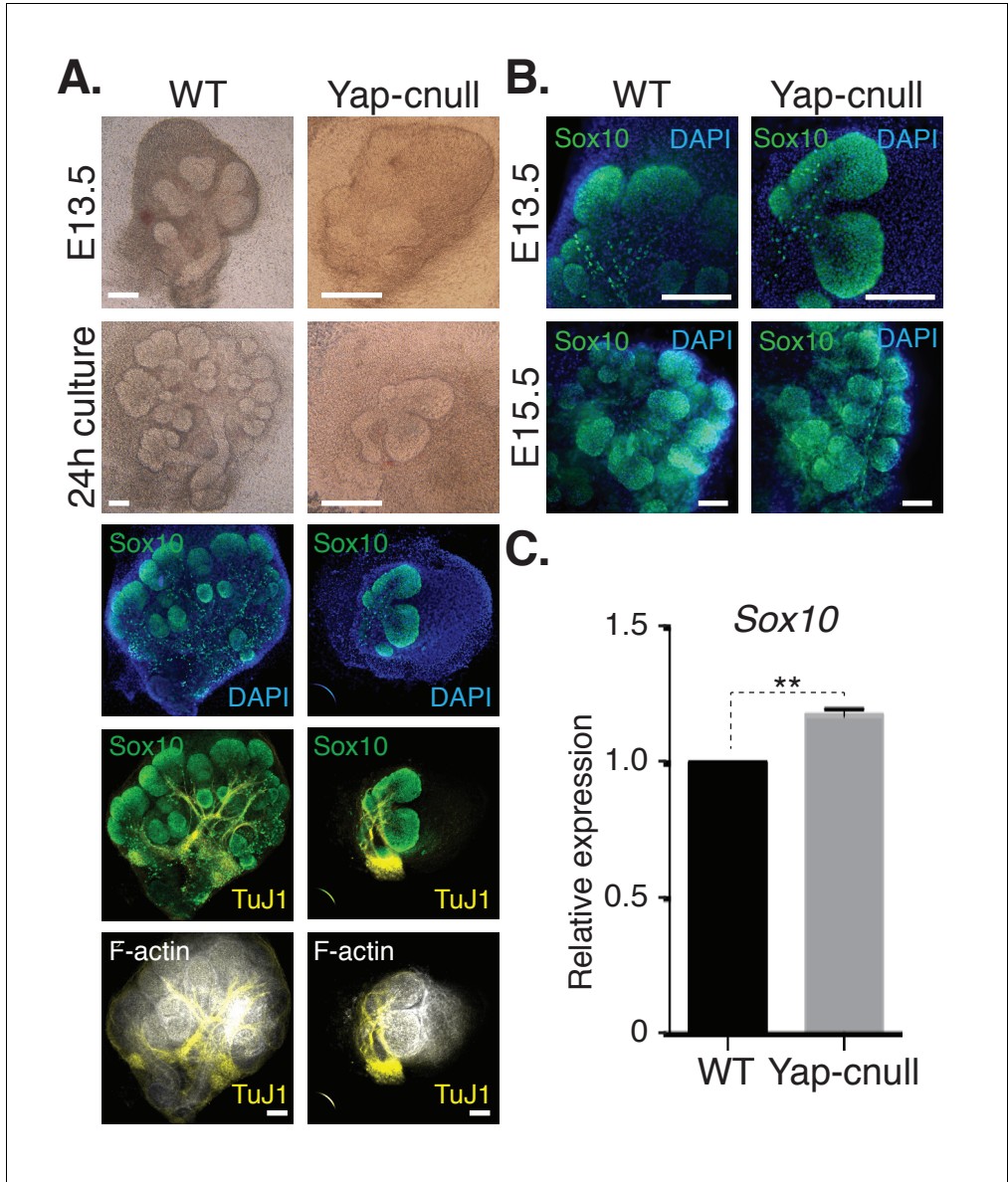

**Figure 4.** A relative increase in the number of cells expressing the bud marker Sox10 is observed in *Yap*-cnull SMGs. E13.5 WT and *Yap*-cnull SMGs were dissected and cultured for 24 hr ex vivo and then examined by microscopy. (**A**) Phase-contrast images were taken of E13.5 WT and *Yap*-cnull SMGs at the time of dissection and 24 hr after culture. The same SMGs (each column is one SMG) were analyzed by IF for Sox10 (green), TuJ1 (yellow), and F-actin (white). DAPI was used to mark the nuclei (blue). Scale = 100 μm. (**B**) Images from IF microscopy analysis of Sox10 in E13.5 (zoomed in from (**A**)) and E15.5 WT and Yap-cnull SMGs. DAPI was used to mark the nuclei (blue). Scale = 100 μm. (**C**) qPCR analysis of *Sox10* expression in E15.5 WT and Yap-cnull SMGs. The average of three experiments is shown +S.E.M. [one sample t-test: **p<0.001]. All images represent observations made from a minimum of three biological repeats.

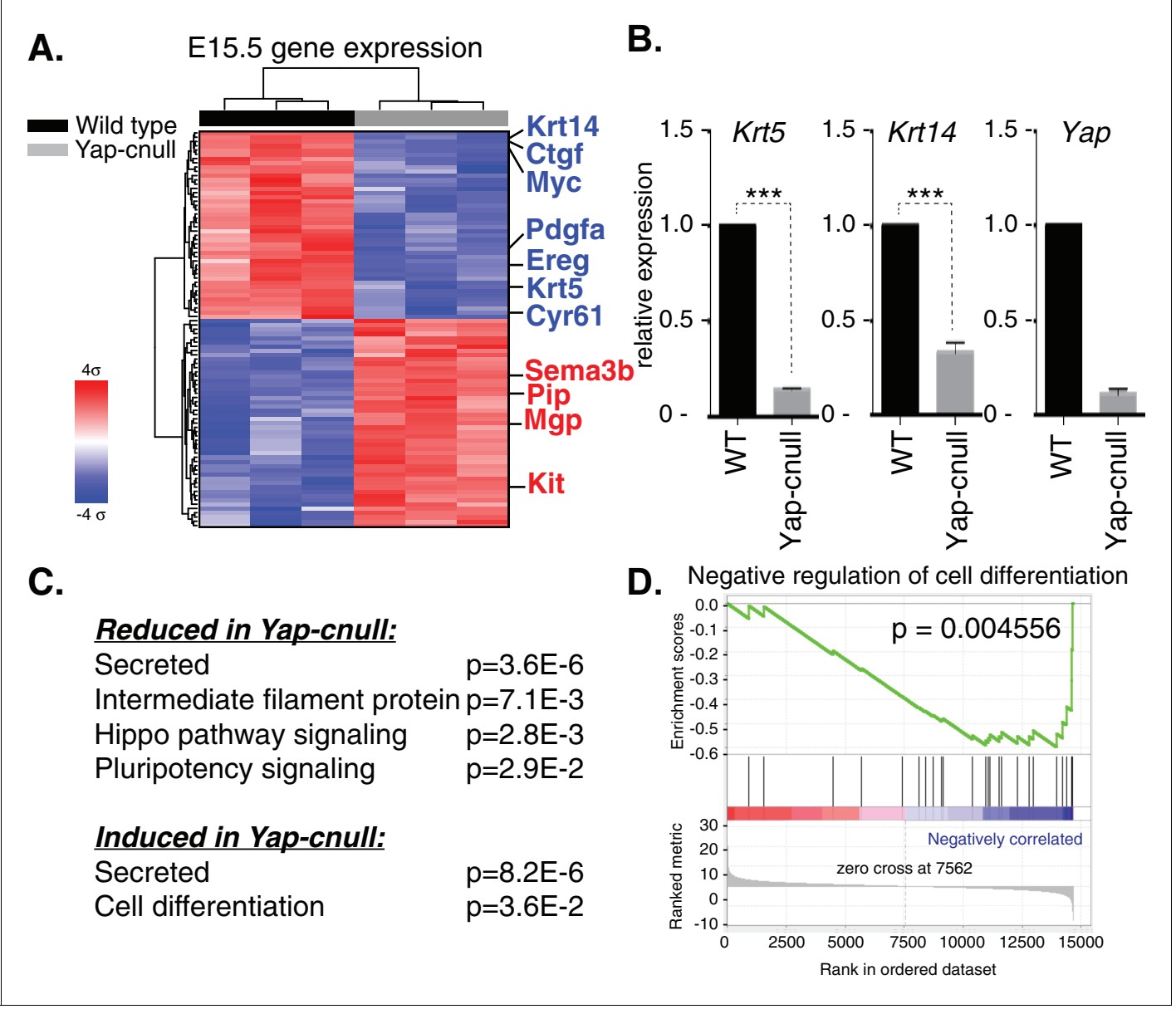

**Figure 5.** Global gene expression analysis of *Yap*-cnull SMGs indicates the aberrant regulation of genes encoding secreted factors and cell fate regulators. (A) Cluster analysis of microarray-generated gene expression data from E15.5 WT vs. *Yap*-cnull SMGs showing genes with a > 2-fold-change and FDR q (filtered) <0.01. Red depicts increased and blue depicts decreased gene expression. Relevant genes are highlighted in blue including canonical Yap targets (*Ctgf*, *Cyr61*) as well as relevant SMG epithelial markers (*Krt14*, *Krt5*, and *Kit*). (B) qPCR analysis of *Yap*, *Krt14*, and *Krt5* expression in E15.5 WT and *Yap*-cnull SMGs. The average of three SMGs from different litters is shown +S.E.M. (one sample t-test: ***p<0.0001). (C) DAVID pathway analysis of genes in (A) that are reduced and induced in *Yap*-cnull SMGs. (D) GSEA of significantly downregulated genes in *Yap*-cnull SMGs shows enrichment for negative regulation of cell differentiation.

The following source data is available for figure 5:

**Source data 1.** Genes differentially expressed in Yap-cnull vs WT (FDRq <0.01; fold change >2).

**Source data 2.** Gene Set Enrichment Analysis (GSEA) of genes differentially expressed in Yap-null versus WT SMGs.

We next interrogated genes differentially expressed in *Yap*-cnull SMGs for growth factors that may potentially be important for altering the microenvironment that directs ductal progenitor specification. Epiregulin (Ereg), an ErbB receptor ligand that has been implicated in cell fate control in other contexts (*Gregorieff et al., 2015*), was significantly repressed in *Yap*-cnull SMGs, which we confirmed by qPCR (*Figure 6A*). To gain insight into the relevance of *Ereg* expression, we examined developing SMGs using RNA in situ analysis, which showed prominent *Ereg* expression in the

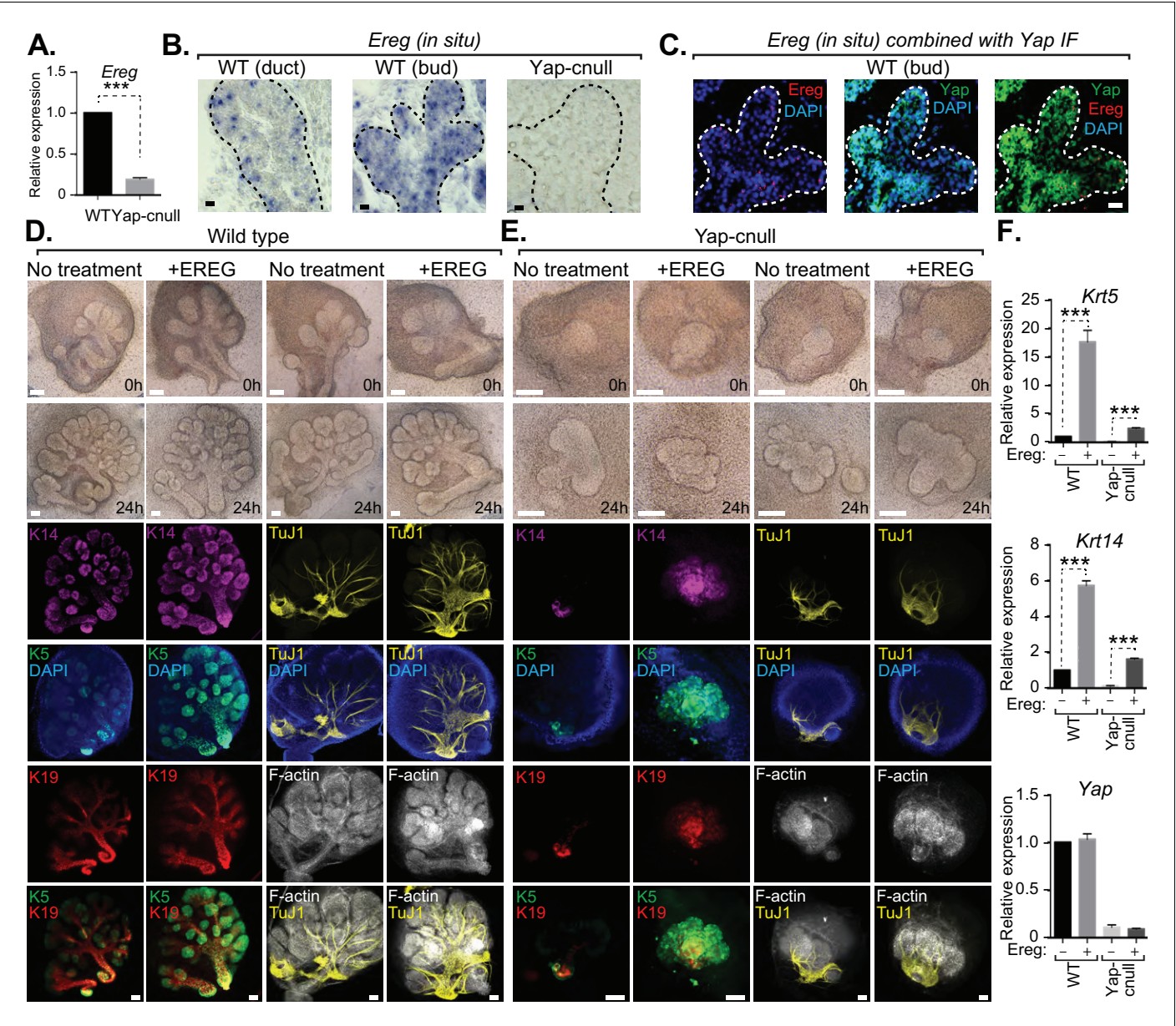

**Figure 6.** Yap-induced Epiregulin (Ereg) expression directs ductal progenitor specification. (A) qPCR analysis of *Ereg* expression in E15.5 WT vs. *Yap*-cnull SMGs. The average of three SMGs from different litters is shown +S.E.M. [one sample t-test: \*\*\*p<0.0001]. (B) In situ hybridization of *Ereg* mRNA in E13.5 WT (duct and bud) and *Yap*-cnull (bud) SMGs. (C) Combined in situ hybridization for *Ereg* mRNA (red) and IF for Yap (green) in E13.5 WT SMGs. (D,E) E13.5 WT and *Yap*-cnull SMGs were dissected and cultured for 24 hr in the presence or absence of 0.5 μg/mL of exogenous Ereg protein and analyzed by phase-contrast and IF for Krt14 (K14, magenta), Krt5 (K5, green), Krt19 (K19, red), TuJ1 (yellow), and F-actin (white, Phalloidin). DAPI was used to mark the nuclei (blue). Scale = 100 μm. (F) qPCR analysis of *Yap*, *Krt5*, and *Krt14* expression in the conditions of (D) and (E). The average of three SMGs from different litters is shown +S.E.M. [one sample t-test: \*\*\*p<0.0001]. All images represent observations made from a minimum of three biological repeats.

epithelium of buds, particularly in the developing ductal progenitor regions, as well as in distinct populations of basal cells of the ductal epithelium (*Figure 6B*). *Ereg* expression levels, however, were completely lost in *Yap*-cnull SMG (*Figure 6B*). Interestingly, co-analysis of *Ereg* levels (RNA in situ) and Yap localization (IF) indicated that cells exhibiting high levels of nuclear Yap also express *Ereg*, suggesting that nuclear Yap activity promotes *Ereg* expression, consistent with our microarray analysis (*Figure 6C*).

To test whether Ereg plays roles in SMG patterning, we treated WT SMG explants with purified exogenous Ereg for 24 hr in ex vivo culture conditions. Ereg treatment led to a morphological thickening and enlargement of the ductal domain that was accompanied by a reduction in the bud domain (*Figure 6D*). IF microscopy analysis showed a striking enrichment in Krt5- and Krt14-positive cells, with these cells composing almost the entire SMG epithelium, including distal epithelial regions (*Figure 6D*). Comparative staining with Tuj1 showed a more broadly distributed nerve in the Ereg-treated glands, and F-actin cytoskeletal analysis indicated that along with the expansion of ductal progenitors in response to Ereg, the cytoskeletal organization was disrupted (*Figure 6D*). Interestingly, treatment of *Yap*-cnull SMG explants with exogenous Ereg partially rescued some of the observed defects. Most notably, we observed the Ereg treatment led to the emergence of Krt5/Krt14-positive cells, with a subset of these cells also expressing Krt19 (*Figure 6E*), all of which were normally absent in *Yap*-cnull SMGs. Analysis of the F-actin cytoskeleton of Ereg-treated *Yap*-cnull SMGs suggested a partial rescue in the organization of duct and bud domains, as some regions exhibited ductal-like F-actin organization and buds showed a slight clefting morphology (*Figure 6E*). Moreover, slight broadening of nerve innervation was also apparent in the Ereg-treated *Yap*-cnull SMGs. To quantify the effects on cell fate, we performed qPCR analysis on parallel SMG explant cultures, and found that treatment of WT and Yap-cnull SMGs with exogenous Ereg led to increased *Krt5* and *Krt14* expression comparable to control glands without treatment (*Figure 6F*). These data indicated that Ereg stimulation of SMG epithelium was capable of rescuing some of the patterning defects observed in *Yap*-cnull epithelium, and that this modestly rescued the associated morphogenesis defects.

To explore the role(s) of endogenous Ereg in SMG epithelial patterning, we transfected E13.5 WT SMGs with either control siRNA or siRNA targeting Ereg, and then cultured the developing SMGs ex vivo for 24 hr. Analysis of RNA isolated from Ereg-depleted SMGs by qPCR indicated a striking deficiency of *Krt5* and *Krt14* expression, accompanied by a small but significant reduction in Yap expression (*Figure 7A*). Microscopy analysis of Ereg-depleted SMGs showed branching defects along with a dramatic loss of Krt5-and Krt14-positive cells (*Figure 7B*), as well a reduction in cells marked with Krt19 (*Figure 7B*). Depletion of Ereg also resulted in diminished nerve innervation and growth (*Figure 7B*). Collectively, these observations offer evidence suggesting Ereg is required for the maintenance of a ductal progenitor niche.

## Loss of Lats1/2-mediated control of Yap in SMG epithelium leads to severe branching morphogenesis defects and uncontrolled ductal epithelial expansion

The Hippo pathway is the primary signaling pathway that controls Yap localization and activity in development and has been implicated in the control of branching morphogenesis (*Reginensi et al., 2016*). The Lats1 and Lats2 kinases (Lats1/2) are known to phosphorylate Yap, which restricts nuclear Yap localization and activity (*Meng et al., 2016*). Given that we observed Yap phosphorylation that correlated with its cytoplasmic localization in maturing SMG ductal epithelium, we next set out to characterize the potential relationship with the Lats1/2 kinases. We started by examining Lats1/2 activity and localization in E15.5 SMGs by IF microscopy using an antibody that recognizes the phosphorylated-active forms of Lats1/2 (p-Lats1/2) (*Szymaniak et al., 2015*). We found that active Lats1/2 was absent in developing bud epithelium and abundant in mature luminal ductal SMG cells (*Figure 8A*). P-Lats1/2 was localized to the apical domain of the majority of luminal ductal cells, correlating with cells that exhibit cytoplasmic Yap localization. We have shown previously that inhibition of Lats2 with siRNA in E13.5 SMG explant cultures gives rise to branching defects (*Enger et al., 2013*), suggesting that p-Lats1/2 in the ductal epithelium plays an important role. To define how Lats1/2 activity impacts SMG development, we used the *Shh*-Cre recombinase to conditionally delete the *Lats1* and the *Lats2* genes in the epithelium of developing mouse SMGs (herein referred to as *Lats1/2*-cnull) (*Heallen et al., 2011*). Deletion of either *Lats1* or *Lats2* alone did not show any

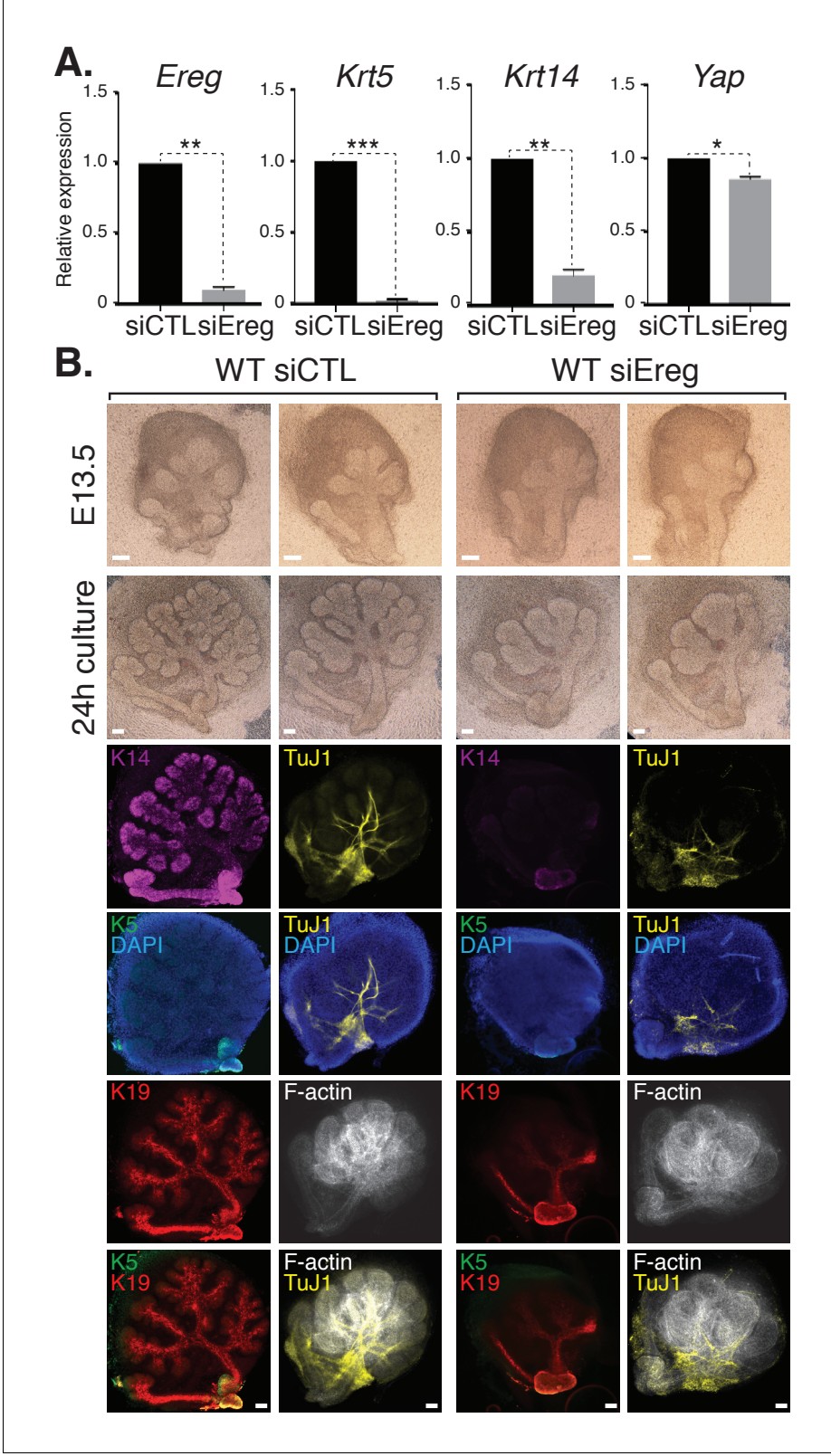

**Figure 7.** Epiregulin knockdown results in the loss of Krt5- and Krt14-positive ductal progenitors accompanied by a disruption of SMG branching. (A) qPCR analysis of *Ereg, Krt5, Krt14,* and *Yap* expression in E13.5 WT SMGs treated with control siRNA or siRNA targeting Ereg. The average of three SMGs from different litters is shown +S. E.M. [one sample t-test: *p<0.01, **p<0.001, ***p<0.0001]. (B) E13.5 WT SMGs were dissected and cultured for 24

*Figure 7 continued on next page*

*Figure 7 continued*

hr in the presence of control siRNA or Epiregulin siRNA and analyzed by phase-contrast and IF for Krt14 (K14, magenta), Krt5 (K5, green), Krt19 (K19, red), TuJ1 (yellow), and F-actin (white, Phalloidin). DAPI was used to mark the nuclei (blue). Scale = 100 μm. All images represent observations made from a minimum of three biological repeats.

observable morphological defects in SMG development (data not shown). However, deletion of both *Lats1* and *Lats2* led to severe morphogenesis defects, with an almost complete lack of branching at early developmental time points (*Figure 8B–D*). E13.5 *Lats1/2*-cnull SMGs displayed an enormous augmentation of the ductal domain at the expense of the distal bud, which continued to expand to a very large size in ex vivo cultures. Similarly, E15.5 *Lats1/2*-cnull SMGs exhibited a large ductal structure with a severely enlarged major/primary ductal structure (*Figure 8D*). IF microscopy analysis showed that almost all the epithelial cells within *Lats1/2*-cnull SMGs exhibited strong nuclear Yap localization and that pS112-Yap levels were completely absent (*Figure 8E*). To investigate the effect of *Lats1/2* deletion on the entire ductal tree, we analyzed E18.5 WT and *Lats1/2*-cnull SMGs for Krt5, Krt14, and Yap by IF microscopy. While WT ducts only express these markers in basal cells, the major ducts in the *Lats1/2*-cnull SMGs were hyperplastic and did not form an observable lumen, with all cells staining positive for Krt5 and Krt14 (*Figure 8F*). Minor ducts in the *Lats1/2*-cnull SMGs were markedly larger and did not exhibit stereotypical branching and budding features as compared to the WT (*Figure 8G*). The minor ducts in *Lats1/2* SMGs were composed predominantly of Krt5/Krt14-positive cells, while equivalent WT regions showed positive staining in few basal-positioned cells (*Figure 8G*). Gene expression analysis of RNA isolated from *Lats1/2*-cnull SMGs confirmed these large increases in *Krt5* and *Krt14* expression (*Figure 8H*).

## EGFR inhibition blunts the patterning defects observed in Lats1/2-cnull SMG epithelium

Our observations suggested that inappropriate nuclear Yap activation resulting from *Lats1/2* deletion promotes the expansion of Krt5/Krt14-positive ductal SMG progenitors. We therefore examined whether Ereg levels were altered in *Lats1/2*-cnull SMGs and observed very high expression of *Ereg* compared to WT SMGs, as measured by qPCR (*Figure 9A*) and RNA in situ hybridization (*Figure 9B*). To examine whether the expansion of cells expressing ductal epithelial progenitor markers within *Lats1/2*-cnull SMGs relied on an Ereg-mediated mechanism, we tested the effects of EGFR inhibition in wild type and *Lats1/2*-cnull developing SMGs ex vivo. Wild type SMGs treated with the EGFR inhibitor AG1478 exhibited branching morphogenesis defects, which were consistent with observations from EGFR-deleted mice (*Häärä et al., 2009*; *Jaskoll and Melnick, 1999*). IF analysis of the EGFR-inhibited SMGs showed diminished patterning of Krt5 and Krt14 cells, reduced levels of mature Krt19 cells, as well as defective organization of the F-actin cytoskeleton and parasympathetic innervation (*Figure 9C*), all of which are phenotypes that resembled *Yap*-cnull SMGs. *Lats1/2*-cnull SMGs grown ex vivo for 24 hr showed an expansion of ductal structures composed of Krt5 and Krt14-positive cells, with a remarkable enlargement of the major ductal region (*Figure 9D*). *Lats1/2*-cnull glands also exhibited defective nerve innervation and F-actin cytoskeletal organization, indicating severely compromised patterning and extracellular signaling (*Figure 9D*). Notably, some cells within *Lats1/2*-cnull SMGs also expressed Krt19, many of which were also positive for Krt5 and Krt14, suggesting that these cells may be in an aberrant primitive state normally not observed in wild type SMGs (*Figure 9D*). Treatment of *Lats1/2*-cnull SMGs with the EGFR inhibitor AG1478 led to a loss of Krt5 and Krt14-positive cells, indicating that EGFR activation drives the expansion of these ductal progenitor-like cells (*Figure 9D*). Further, gene expression analysis by qPCR validated the observed differences in *Krt5* and *Krt14* expression in our IF microscopy experiments, with AG1478 treatment of either WT and *Lats1/2*-cnull SMGs leading to an almost complete loss of the expression of these ductal progenitor markers (*Figure 9E*).

Collectively, our observations suggest that nuclear Yap plays an essential role in the development of ductal epithelial SMG progenitors (see model in *Figure 10*) and that Lats1/2-mediated removal of Yap from the nucleus is required for the maturation of ductal structures.

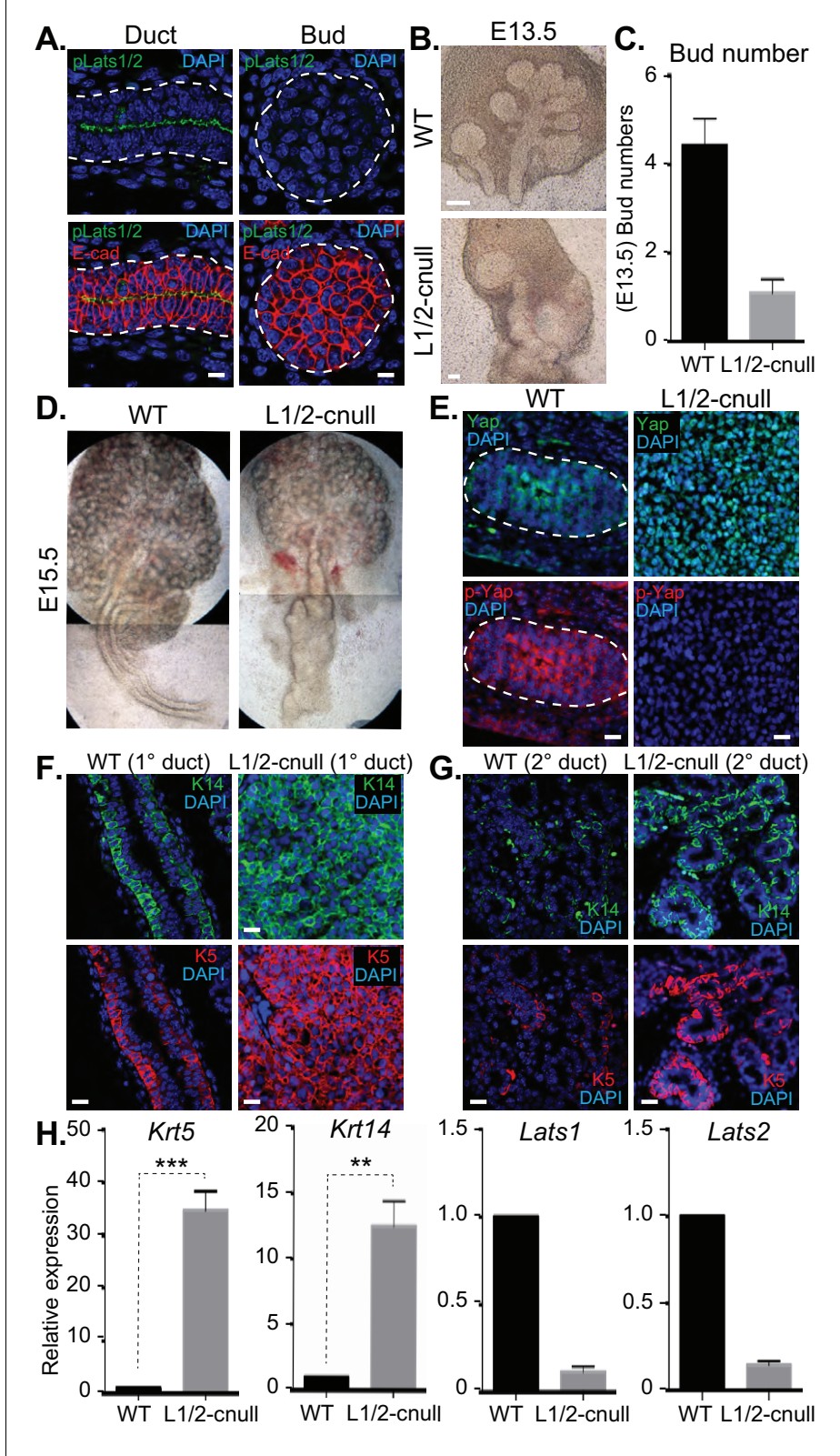

**Figure 8.** Deletion of Lats1/2 in developing SMG epithelium leads to aberrant nuclear Yap localization and severe branching morphogenesis and patterning defects. (**A**) IF microscopy analysis of E15.5 SMGs for phospho-Lats1/2 (green) and E-cadherin (red), indicating active Lats1/2 in luminal cells of the ductal epithelium. Scale = 10 μm. (**B**) Phase-contrast images of E13.5 WT and *Shh*-Cre-*Lats1/2* null (*Lats1/2*-cnull) SMGs. Scale = 100 μm (**C**)

*Figure 8 continued on next page*

*Figure 8 continued*

Quantitation of bud number from E13.5 WT and *Lats1/2*-cnull SMGs. n = 21. (D) Phase-contrast images of E15.5 WT and *Lats1/2*-cnull SMGs highlighting the severe ductal expansion phenotype. Note that each respective image is stitched together from two images. (E) Images from IF microscopy analysis of E15.5 WT and *Lats1/2*-cnull SMGs for Yap (green), phospho-S112 Yap (red), and DAPI (blue). Scale = 10 μm. (F) Images from IF microscopy analysis of E15.5 WT and *Lats1/2*-cnull primary/major (1°) ducts for Krt14 (K14, green) and Krt5 (K5, red). Scale = 10 μm. (G) Images from IF microscopy analysis of E15.5 WT and *Lats1/2*-cnull minor (2°) ducts for Krt14 (K14, green) and Krt5 (K5, red). Scale = 10 μm. (H) qPCR analysis of *Lats1*, *Lats2*, *Krt5*, and *Krt14* expression in E15.5 WT vs. *Lats1/2*-cnull SMGs. The average of three experiments is shown +S.E.M. [one sample t-test: **p<0.001; ***p<0.0001]. All images represent observations made from a minimum of three biological repeats.

## Discussion

We present data describing an essential role for the Hippo pathway effector Yap in the development of the salivary gland. We show that the deletion of Yap in developing SMG epithelium leads to severe patterning and morphogenesis defects. Our data suggest that these defects arise, in large part, from the loss of nuclear Yap transcriptional activity, which defines signals important for instructing the specification of ductal epithelial progenitors. This conclusion is based on our observations

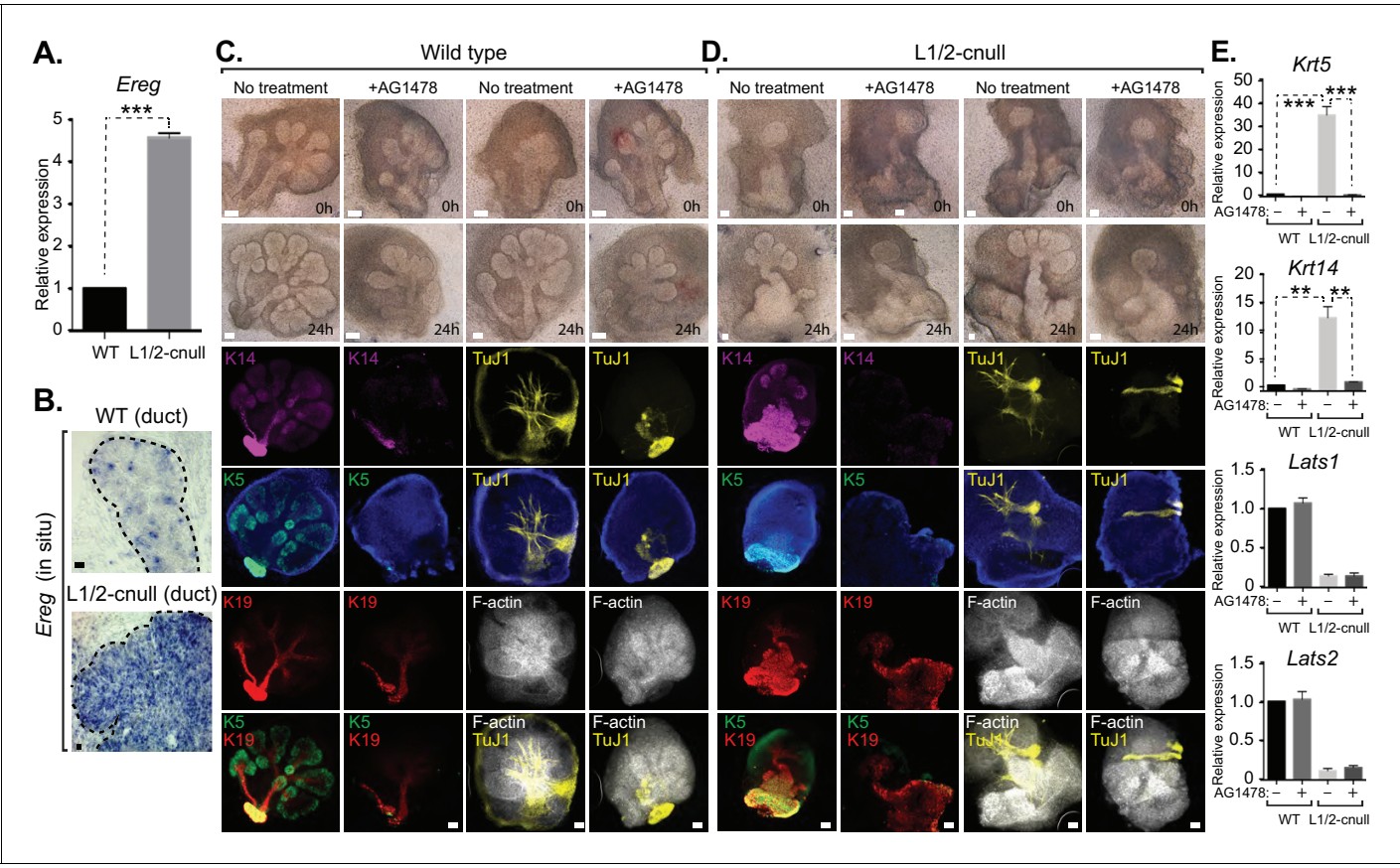

**Figure 9.** EGFR inhibition blunts the patterning defects observed in *Lats1/2*-cnull SMG epithelium. (A) qPCR validation of *Ereg* expression levels in E15.5 WT vs. *Lats1/2*-cnull SMGs. The average of three experiments is shown +S.E.M. [one sample t-test: ***p<0.0001]. (B) In situ hybridization of *Ereg* mRNA in E13.5 WT and *Lats1/2*-cnull SMGs. (C,D) E13.5 WT and *Lats1/2*-cnull SMGs were dissected and cultured for 24 hr in the presence or absence of 10 μM EGFR inhibitor AG-1478 and analyzed by phase-contrast and IF for Krt14 (K14, magenta), Krt5 (K5, green), Krt19 (K19, red), TuJ1 (yellow), and F-actin (white, Phalloidin). DAPI was used to mark the nuclei (blue). Scale = 100 μm. (E) qPCR analysis of *Lats1*, *Lats2*, *Krt14*, and *Krt5* expression in the conditions of (C) and (D). The average of three experiments is shown +S.E.M. [one sample t-test: **p<0.001; ***p<0.0001]. All images represent observations made from a minimum of three biological repeats.

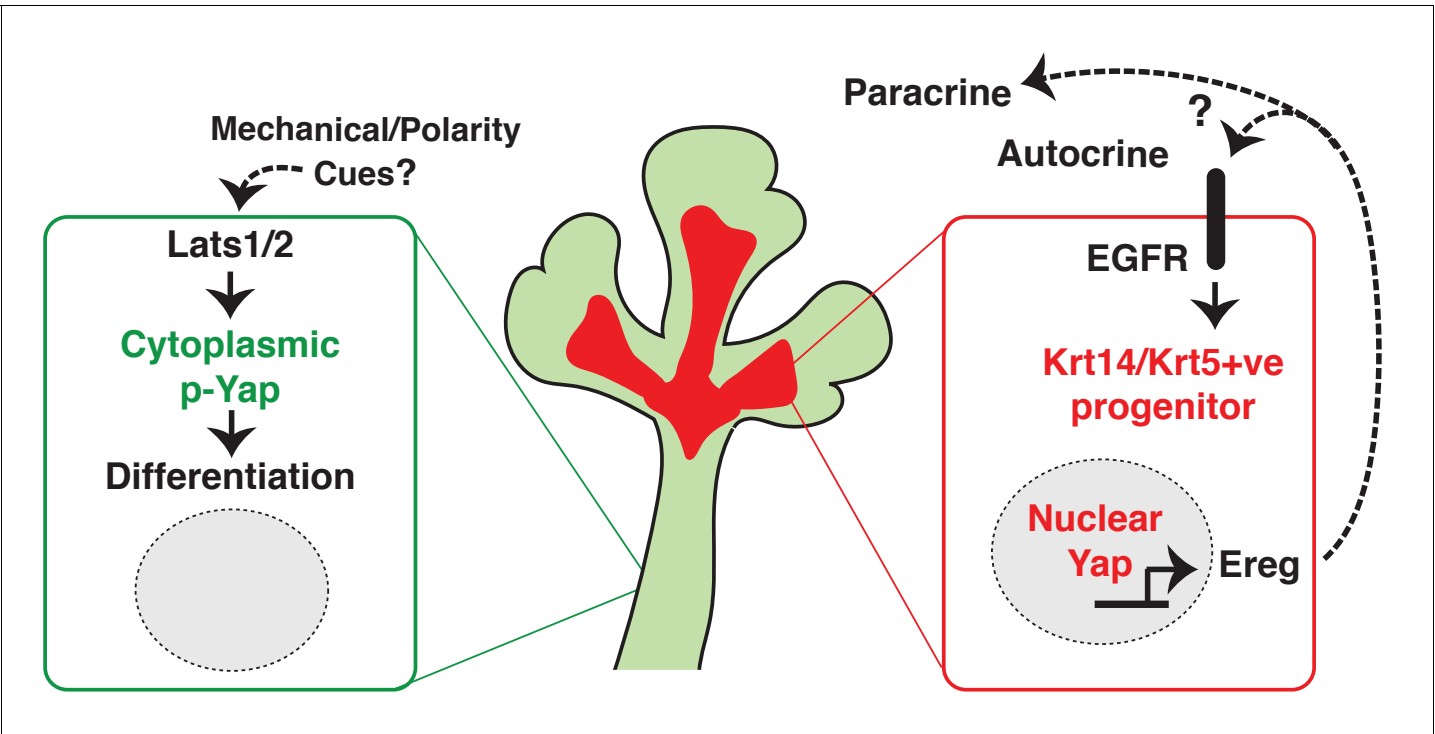

**Figure 10.** Illustration depicting the roles of Yap in submandibular gland epithelial development. We propose nuclear Yap specifies the identity of ductal progenitors, in part by promoting the expression of Ereg and subsequent activation of EGFR signaling. Activated Lats1/2 in the maturing ductal structures promotes the phosphorylation of Yap, which directs the removal of nuclear Yap to support ductal epithelial differentiation.

that *Yap*-cnull SMG epithelium fails to develop ductal domains and shows severely reduced numbers of Krt5 and Krt14-positive cells, which are markers that have been associated with ductal progenitors (*Knox et al., 2010*; *Lombaert et al., 2013*). We show that developing SMG epithelium displays distinct localization patterns for Yap in various regions, with nuclear Yap most pronounced in the cells of the developing bud, cells developing into ductal areas, as well as in subpopulations of basal epithelial cells that line the SMG ducts, which interestingly are all regions that exhibit cytoskeletal tension (*Harunaga et al., 2011*). In contrast, cytoplasmic Yap is observed in regions associated with luminal differentiation of the ducts, suggesting that the removal of nuclear Yap is required for SMG epithelial differentiation. Consistent with this premise, deletion of the Hippo pathway Lats1/2 kinases in developing SMG epithelium resulted in aberrant nuclear Yap localization and expansion of Krt5 and Krt14-positive cells in regions that normally undergo ductal maturation.

Hippo pathway-mediated restriction of nuclear Yap activity has classically been associated with the governance of organ size via a growth-restricting mechanism that relies on precise coordination between proliferation and apoptosis (*Pan, 2010*). As such, Yap and upstream Hippo pathway effectors have emerged as influential oncogenes and tumor-suppressors, respectively. As evidenced here, dynamic Hippo signaling during epithelial morphogenesis extends beyond the view that Hippo signaling controls organ size only by directing cell growth. Rather, our observations from *Yap*- and *Lats1/2*-cnull SMGs indicate that Hippo pathway activity is primarily involved in directing a balance in the specification, renewal, and differentiation of ductal progenitors. Similar roles for Hippo pathway-mediated Yap signaling have been described in other branching organs, including the lung, kidney and pancreas (*Gao et al., 2013*; *George et al., 2012*; *Mahoney et al., 2014*; *Reginensi et al., 2013*). Interestingly, however, although Yap plays key roles in progenitor control across different organs, the mechanism(s) by which Yap exerts these functions appears to be context-dependent, likely relating to the distinct signaling network that is organized in each respective organ.

The identification of a role for Ereg in SMG patterning and morphogenesis resembles its role in other systems. In the intestine, a Yap-Ereg network that involves neighboring stromal cells is central

for promoting epithelial cell survival and inducing a regenerative program (*Gregorieff et al., 2015*). However, unlike the intestine, our data suggest that in the SMG, Yap promotes the expression of Ereg in the epithelium to sustain a progenitor niche. Accordingly, we observed elevated Ereg expression in the newly forming ductal progenitor regions and in distinct regions of basal ductal epithelium, precisely the sites with the highest levels of nuclear Yap. We further found that depletion of Ereg or small molecule-mediated inhibition of EGFR, a major receptor for Ereg, results in epithelial branching and maturation defects that resemble those in Yap-cnull SMGs. These observations paralleled those observed following the deletion of EGFR (*Häärä et al., 2009*; *Jaskoll and Melnick, 1999*) and are consistent with recent observations that heparin binding-EGF treatment of SMGs can promote the expansion of Krt5 progenitors (*Knox et al., 2010*). Importantly, we show that ex vivo treatment of developing SMG organ cultures with Ereg expands Krt5 and Krt14-positive cells even in the absence of Yap, indicating that Ereg signaling is sufficient to overcome cell fate defects associated with Yap deletion. Together, these data support a model (illustrated in *Figure 10*) in which nuclear Yap-induced Ereg defines a niche of EGFR signaling that is central in promoting the identity of ductal progenitors.

Signals transduced by the acetylcholine (Ach)/muscarinic (M) receptor 1 in the parasympathetic submandibular ganglion have been shown to increase EGFR protein expression and increase the expansion of Krt5 progenitors in the SMG (*Knox et al., 2010*). Therefore, it is likely that signals emanating from the nerve may crosstalk with Yap, or Yap-regulated signals may communicate with the nerve to control ductal progenitor specification and maturation. Consistent with this idea, we observed diminished and disorganized parasympathetic innervation in both the Yap-cnull and Lats1/2-cnull SMGs. Recent work has shown that Neuregulin1 (Nrg1), a neuregulin family ligand that can activate EGFR, promotes the expression of Wnt ligands that signal to direct parasympathetic innervation (*Knosp et al., 2015*). The Hippo signaling pathway is highly interconnected with the Wnt pathway (*Azzolin et al., 2014*; *Varelas et al., 2010a*; *Heallen et al., 2011*). Thus, it is tempting to speculate that Yap-nerve signaling crosstalk plays a key role in ductal progenitor patterning and may be linked to signals that direct epithelial branching.

Our observations implicate the dynamics of Yap localization in the control of ductal epithelial maturation. Several studies have shown that the developing ductal epithelium acquires specialized polarity cues as it matures. Given that polarity cues are tightly integrated with the control of Yap localization and activity (*Szymaniak et al., 2015*; *Varelas et al., 2010b*), it is likely that epithelial polarity-mediated regulation of Yap directs the differentiation of the ductal epithelium. Indeed, recent work in the developing lung epithelium shows that polarity-regulating proteins, such as the Crumbs transmembrane proteins that direct apical domain specification, promote interactions between the Lats1/2 kinases and Yap (*Szymaniak et al., 2015*). The dynamics of these polarity proteins, which may be controlled by the mechanical microenvironment, direct the localization of Yap and control cell differentiation. For example, basal stem cell cells in the lung epithelium lack aspects of epithelial polarity and exhibit high levels of nuclear Yap that promotes the basal stem cell identity (*Szymaniak et al., 2015*; *Zhao et al., 2014*).

Notably, we observed a subset of basal epithelial cells in maturing SMG ducts with prominent nuclear Yap localization. These basal cells, a subset of which are marked by Krt5 in adult SMGs, are thought to possess stem cell activity that can repair adult SMG epithelium upon damage (*Knox et al., 2013*). Therefore, nuclear Yap may contribute to the identity of these stem cells as it does in other organs, such as the skin and the lung (*Silvis et al., 2011*; *Szymaniak et al., 2015*; *Zhao et al., 2014*). However, nuclear Yap in the ductal basal cells did not perfectly correlate with Krt5 expression, suggesting that the Krt5-positive population is composed of distinct subpopulations, as proposed previously (*Lombaert and Hoffman, 2010*), or that Yap localization is dynamic in these cells and our observations captured only a snapshot. Our in situ analysis of *Ereg* expression in maturing ducts also suggested that only a sub-population of basal ductal cells express *Ereg*. Thus, it is possible that a Yap-Ereg signaling niche specifies a unique basal progenitor identity. Such a progenitor niche may be dysregulated in salivary gland carcinomas, as Krt5-positive populations are frequently amplified in these tumors. Many studies have implicated Yap as a pro-tumorigenic factor (*Harvey et al., 2013*), and thus knowledge into the developmental mechanisms, such as that provided by our study, may offer insight into the etiology of salivary gland diseases. One such disease may be Sjogren's Syndrome, a debilitating autoimmune disease that affects salivary secretion, as aberrant Yap localization has been observed in the salivary gland epithelium of patients

(*Enger et al., 2013*). Similarly, Krt5-expressing cell populations have been observed to be expanded in Sjogren's Syndrome epithelium (*Gervais et al., 2015*). Thus, dysregulated Yap-mediated cell fate control may be linked to this diseased state.

In summary, our observations indicate that Yap is required for directing ductal SMG epithelial progenitor patterning, with nuclear Yap inducing the expression of Ereg, which drives signals for the specification of ductal epithelial progenitors; removal of Yap from the nucleus is therefore a requirement for the maturation of ductal structures. Importantly, our work highlights similarities in Yap signaling with other branching organs while also uncovering significant differences. Thus, given the importance of Yap signaling in organ development and disease, understanding this context is an important challenge for the future.

## Materials and methods

### Mice

Developmental studies on wild-type mice were performed using C57BL/6 mice. The *Yap*-loxP/loxP mice (*Reginensi et al., 2013*) were generously provided by Dr. Jeff Wrana (LTRI, Toronto, CN) in an ICR/S129 mixed background and bred with C57BL/6 *Shh*-gfpcre (Jackson Laboratories, #005622) mice to generate *Yap*-cnull animals. The *Lats1*-loxP/loxP and *Lats2*-loxP/loxP mice (*Heallen et al., 2013*; *Heallen et al., 2011*) were generously provided by Dr. Randy Johnson (MD Anderson, Houston, TX) and bred to the *Shh*-gfpcre mice to generate *Lats1/2*-cnull animals. All animal experiments were done in accordance with protocols approved by the Institutional Animal Care and Use Committee at Boston University.

### Ex vivo SMG organ cultures

SMG explants were cultured as previously described (*Steinberg et al., 2005*). Briefly, freshly dissected E13.5 SMGs were plated on nucleopore filters for 24 hr and were either lysed for RNA or fixed for immunostaining. When indicated, explants were treated with DMSO or PBS as a control, 0.5 µg/mL Epiregulin (Fisher; 1068EP050) or 10 µM AG1478 (Sigma; T4182). For Ereg knockdown, freshly dissected E13.5 SMGs were transfected with siRNA targeting murine Ereg (CTCAAG TGCAGATTACAAA) or control siRNA (Qiagen, 1027310) delivered by Lipofectamine RNAiMax transfection reagent (Thermo Scientific, 13778030) and cultured for 24 hr before processing.

### Microarray analysis

E15.5 WT and *Yap*-cnull SMGs were dissected and the epithelium was separated from the mesenchyme as described (*Rebustini and Hoffman, 2009*). RNA was extracted from the epithelium using either the RNeasy kit or the miRNeasy Micro kit (QIAGEN). The amount of isolated RNA was normalized and automated sample amplification and biotin labeling were carried out using the NuGEN Ovation RNA Amplification system V2, Ovation WB reagent and Encore Biotin module according to manufacturer's recommendations using an Arrayplex automated liquid handler (Beckman Coulter). Two micrograms of biotin-labeled sscDNA probe were hybridized to a HT_MG-430_PM Affymetrix Array Plate with modified conditions as previously described (*Allaire et al., 2013*). Washing and staining of the hybridized arrays were completed as described in the GeneChip Expression analysis technical manual for HT plate arrays using the Genechip Array Station (Affymetrix) with modifications as previously described (*Allaire et al., 2013*). The processed GeneChip plate arrays were scanned using a GeneTitan scanner (Affymetrix). Raw CEL files (available at Gene Expression Omnibus (GEO) Series GSE90480) were normalized to produce gene-level expression values using the implementation of the Robust Multiarray Average (RMA) in the *affy* R package (version 1.36.1) and an Entrez-Gene-specific probeset mapping (17.0.0) from the Molecular and Behavioral Neuroscience Institute (Brainarray) at the University of Michigan. Differential expression was assessed using the moderated *t* test implemented in the *limma* R package (version 3.14.4). Correction for multiple hypothesis testing was accomplished using the Benjamini-Hochberg false discovery rate (FDR). Human homologs of mouse genes were identified using HomoloGene (version 68). All microarray analyses were performed using the R environment for statistical computing (version 2.15.1). GSEA (version 2.2.1) was performed in a pre-ranked manner (default parameters with random seed 1234) using a list of Entrez

Gene identifiers of the human homologs of the genes interrogated by the array, ranked according to the *t* statistic computed between the Yap-cnull and wild-type groups. Mouse genes with multiple human homologs (or *vice versa*) were removed prior to ranking, so that the ranked list represents only those human genes that match exactly one mouse gene. Biocarta, KEGG, Reactome, Gene Ontology (GO), and transcription factor and microRNA motif gene sets (in human Entrez Gene ID space) were obtained from the Molecular Signatures Database (MSigDB), version 5.0.

## qPCR

For quantitative real-time PCR, RNA was extracted using the RNeasy kit or miRNeasy Micro Kit (QIA-GEN) and reverse-transcribed using iScript enzymes (BioRad). Reactions were prepared using Fast SYBR Green Master Mix (Applied Biosystems; 4385612) and carried out in a ViiA7 Real-Time PCR System (Applied Biosystems) and analyzed using the ddCT method. Primer sequences are listed in the *Table 1*. Statistics were performed using Prism 7 (Graphpad).

## Immunofluorescence microscopy

Embryonic SMGs were dissected and fixed in 4% PFA (Electron Microscopy Sciences; 15710) for 15 min to an hour and processed for paraffin embedding. Staining was performed using a standard dewaxing and hydration protocol, followed by a microwave-assisted antigen retrieval step using a low-pH buffer (Vector Labs; H-3300). Primary antibodies and dilutions used are described in *Table 2*. Secondary antibodies used were conjugated to Alexa Fluor-488,−555, −568,−594, or −647 fluorophores (Life Technologies). Slides were mounted using ProLong Gold Antifade Reagent (Life Technologies; P36930), and images were captured using a confocal microscope system (Carl Zeiss; LSM710) or an inverted epi-fluorescent microscope (Carl Zeiss; Axio Observer.D1).

## In situ hybridization

In situ hybridization was performed as previously described (*Mahoney et al., 2014*) and adapted for paraffin-embedded tissue slides. The DIG-labeled *Ereg* probe (*Gregorieff et al., 2015*) was generously supplied by Dr. Jeffrey Wrana (Lunenfeld-Tanenbaum Research Institute, Toronto, Canada). For the combined in situ/IF experiments, hybridization was carried out as above, but DIG was detected using an antibody conjugated to HRP (DIG-POD) (Roche), which was then amplified with tyramide (Perkin Elmer), and visualized with streptavidin-594 (Life Technologies). Primary incubation

**Table 1.** qPCR primer sequences.

| Target | Direction | Sequence | Product size |
|--------|-----------|----------|--------------|
| GAPDH | Forward | TGTTCCTACCCCCAATGTGT | 137 bp |
| GAPDH | Reverse | GGTCCTCAGTGTAGCCCAAG | 137 bp |
| Yap | Forward | AATGTGGACCTTGGCACACT | 106 bp |
| Yap | Reverse | ACTCCACGTCCAAGATTTCG | 106 bp |
| Lats1 | Forward | GCGATGTCTAGCCCATTCTC | 135 bp |
| Lats1 | Reverse | GGTTGTCCCACCAACATTTC | 135 bp |
| Lats2 | Forward | ACAGAGACGCAGCTGAAGGT | 101 bp |
| Lats2 | Reverse | CACAGCTTCGTGATGAGGTC | 101 bp |
| Krt5 | Forward | GGAGCAGATCAAGACCCTCA | 145 bp |
| Krt5 | Reverse | CGGATCCAGGTTCTGCTTTA | 145 bp |
| Krt14 | Forward | AGCGGCAAGAGTGAGATTTCT | 106 bp |
| Krt14 | Reverse | CCTCCAGGTTATTCTCCAGGG | 106 bp |
| Sox10 | Forward | GACCAGTACCCTCACCTCCA | 83 bp |
| Sox10 | Reverse | CGCTTGTCACTTTCGTTCAG | 83 bp |
| Ereg | Forward | TTCTCATCATAACCGCTGGA | 102 bp |
| Ereg | Reverse | CCCCTGAGGTCACTCTCTCA | 102 bp |

**Table 2.** Antibodies used.

| Antigen | Species | Company | Cat# | Dilution | Lot# |
|---|---|---|---|---|---|
| Yap | Mouse | Santa Cruz | 101199 | 1/100 | B2713 and A0512 |
| Yap | Rabbit | CST | D8H1X | 1/100 | 1 |
| Phospho-Yap | Rabbit | CST | 13008S | 1/100 | 1 and 2 |
| Phospho-LATS1/2 | Rabbit | Assay Bio Tech | A8125 | 1/100 | 118125 |
| Krt5 | Chicken | Biolegend | 905901 | 1/300 | D16CF00791 |
| Krt5 | Rabbit | Biolegend | 905501 | 1/300 | D15LF02531 |
| Krt14 | Mouse | abcam | ab7800 | 1/100 | GR185613-1 |
| Krt19 | Rat | DSHB | TROMA-III-c | 1/100 | 11/12/2015 |
| Sox10 | Goat | Santa Cruz | 17342 | 1/100 | F0315 |
| TuJ1 | Mouse | R and D Systems | BAM1195 | 1/500 | HVS0215121 |
| Phalloidin | n/a | Alexa Fluor | A22287 | 1/1000 | 866764 |
| Krt8 | Rat | DSHB | TROMA-1-c | 1/500 | 12/31/2014 |
| Ki-67 | Mouse | BD | 550609 | 1/100 | 67176 |
| Ki-67 | Rabbit | abcam | ab16667 | 1/100 | GR86024-1 |
| PCNA | Mouse | CST | 2586P | 1/100 | 5 |
| Cleaved Caspase 3 | Rabbit | CST | 9661S | 1/100 | 43 |

with the Yap antibody was carried out when the DIG-POD antibody was applied, and incubation with the secondary antibody was carried out before tyramide amplification.

## Whole-mount immunofluorescence

Embryonic SMGs were dissected and fixed in 4% PFA as above or with an ice-cold 1:1 acetone/methanol mixture at −20°C for 15 min, depending on the antibodies to be used. In general, TuJ1/Phalloidin worked in PFA while the keratin antibodies worked in acetone/methanol. SMGs were permeabilized in 0.5% Triton-X 100 (American Bioanalytical; AB02025) for 30 min to an hour at 4°C, and blocked in the same mix plus 10% donkey serum (Fisher; 50-588-37) for up to 2 hr at 4°C. Primary and secondary antibody incubations were in the serum mixture, and either at 4°C overnight or 1 hr at room temperature. After both primary and secondary incubations, SMGs were washed extensively in 0.5% Triton-X 100 (at least 3, up to six changes; 10 min each) and post-fixed with 4% PFA for 30 min. Counterstaining was performed using Hoechst for 15 min. SMGs were immediately imaged by placing them on a coverslip. For increased resolution, SMGs were compressed between two coverslips before imaging, and imaged using an epi-fluorescent microscope (Carl Zeiss; Axio Observer. D1).

## Acknowledgements

We would like to thank Dr. Jeffrey Wrana and Alexander Gregorieff (Lunenfeld Tanenbaum Research Institute, Toronto, Canada) for the *Yap*-loxP/loxP mice (*Reginensi et al., 2013*) for the DIG-labeled *Ereg* probe (*Gregorieff et al., 2015*), and Dr. Randy Johnson (MD Anderson, Houston, TX) for the *Lats1*-loxP/loxP and *Lats2*-loxP/loxP mice (*Heallen et al., 2013*, *2011*). We also thank Norm Allaire and Alice Thai for help with the microarrays, as well as Grant Duclos and the Boston University Clinical and Translational Science Institute (CTSI) for help with data analysis (CTSI grant UL1-TR001430). X. Varelas is funded by research grant no. 1-FY14-219 from the March of Dimes Foundation, by the NIH National Heart Lung and Blood Institute (R01 HL124392) and by funds from the Sjogren's syndrome foundation. M. Kukuruzinska is funded by the NIH National Institute of Dental and Craniofacial Research (R21 DE024954).

## Additional information

### Competing interests

TLR, MM: Employee of Biogen. The other authors declare that no competing interests exist.

### Funding

| Funder | Grant reference number | Author |
|---|---|---|
| March of Dimes Foundation | 1-FY14-219 | Xaralabos Varelas |
| National Heart, Lung, and Blood Institute | R01 HL124392 | Xaralabos Varelas |
| Sjogren's Syndrome Foundation | Research Grant | Xaralabos Varelas |
| National Institute of Dental and Craniofacial Research | R21 DE024954 | Maria Kukuruzinska |
| National Center for Advancing Translational Sciences | UL1-TR001430 | Adam C Gower<br>Xaralabos Varelas |

The funders had no role in study design, data collection and interpretation, or the decision to submit the work for publication.

### Author contributions

ADS, Data curation, Formal analysis, Validation, Investigation, Methodology, Writing—original draft, Writing—review and editing; RM, Data curation, Investigation, Visualization, Methodology; SEM, Data curation, Investigation, Visualization; ACG, Data curation, Formal analysis, Methodology; TLR, MM, Resources, Data curation; MK, Resources, Funding acquisition, Investigation, Methodology, Writing—original draft; XV, Conceptualization, Data curation, Formal analysis, Supervision, Funding acquisition, Investigation, Visualization, Methodology, Writing—original draft, Project administration, Writing—review and editing

### Author ORCIDs

Xaralabos Varelas, http://orcid.org/0000-0002-2882-4541

### Ethics

Animal experimentation: This study was performed in strict accordance with the recommendations in the Guide for the Care and Use of Laboratory Animals of the National Institutes of Health. Animal care and handling was consistent with the recommendations of the Panel on Euthanasia of the American Veterinary Medical Association. Prior to the initiation of experiments, all study protocols were reviewed and modified according to the suggestions of the Boston University School of Medicine IACUC. The Boston University School of Medicine animal management program is accredited by the American Association for the Accreditation of Laboratory Animal Care, and meets National Institutes of Health standards as set forth in the Guide for the Care and Use of Laboratory Animals (DHHS Pub.No. (NIH) 85-23, rev 1985). Boston University's Animal Welfare Assurance number is A-3316-01.

## Additional files

### Major datasets

The following dataset was generated:

| Author(s) | Year | Dataset title | Dataset URL | Database, license, and accessibility information |
|---|---|---|---|---|
| Szymaniak A, Varelas X | 2017 | The Hippo pathway effector YAP is an essential regulator of submandibular gland ductal progenitor patterning | https://www.ncbi.nlm.nih.gov/geo/query/acc.cgi?acc=GSE90480 | Publicly available at the NCBI Gene Expression Omnibus (accession no: GSE90480) |

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
