## [Decision Letter]

[Editors’ note: this article was originally rejected after discussions between the reviewers, but the authors were invited to resubmit after an appeal against the decision.]

Thank you for submitting your work entitled "The Hippo pathway effector YAP is an essential regulator of submandibular gland ductal progenitor patterning" for consideration by *eLife*. Your article has been reviewed by three peer reviewers, one of whom is a member of our Board of Reviewing Editors, and the evaluation has been overseen by a Senior Editor. The reviewers have opted to remain anonymous.

Our decision has been reached after consultation between the reviewers. Based on these discussions and the individual reviews below, we regret to inform you that your work will not be considered further for publication in *eLife*.

The reviewers found your study interesting and highly relevant to the salivary gland, but felt that it lacks the level of novel insight expected for *eLife*. Additionally, there were concerns about extensive reuse of images (for instance, some images appear in Figure 2, Figure 3, Figure 6 and Figure 8) and lack of information on the numbers of experiments performed, throughout the manuscript.

Reviewer #1:

In this manuscript, the authors address the role of Yap and its regulators Lats1 and 2 in development of the submandibular gland (SMG). They show by conditional KO (using *Shh*-Cre) that Yap is needed for normal patterning and morphogenesis of the SMG; deletion of Yap results in loss of ductal structures, both in vivo and in explant culture. To gain insight into the mechanism through which Yap regulates SMG development, they then analysed the transcriptomes of E15.5 WT versus CKO SMG. This indicated Epiregulin as among the genes downregulated in the absence of Yap. Subsequent explant culture experiments showed that Epiregulin could partially compensate for loss of Yap; addition of Epiregulin to the in vitro cultures increased the size of the ductal domain and area expressing Keratins 5, 14 and 19, and from this the authors conclude that Epiregulin expression downstream of Yap regulates expansion of ductal progenitors. The authors then go on to show that regulation of Yap by Lats1 and 2 is required for differentiation of ductal epithelial cells; in the absence of Lats1/2, the primary ductal domain is enlarged and shows nuclear Yap localisation, along with substantially increased Epiregulin expression. By pharmacological inhibition, the authors show that signalling through EGFR (the receptor for Ereg) is required for expansion of the ductal progenitors observed in the absence of Lats1/2. Overall the manuscript is very clearly presented and the data appear to be of a high standard.

Reviewer #2:

In this manuscript Syzmaniak et al. expands on the known roles of Yap in epithelial tissues and progenitor cell regulation by describing a novel role for YAP in regulating epithelial morphogenesis during embryonic salivary gland development. The authors show that genetic ablation of YAP and its regulators LATS1/2 impair epithelial branching morphogenesis and duct formation as well as maintenance of progenitor cells marked by KRT5 and KRT14. Furthermore, they show using genetic experiments and gene expression profiling that YAP is a positive regulator of the EGFR ligand epiregulin: in the absence of epithelial YAP, epiregulin is lost and that epiregulin may be a regulator of the progenitors, thus linking YAP to EGFR signaling. Using ex vivo cultures of the YAP mutant they are able to show that exogenous epiregulin can rescue progenitor cells (if not morphogenesis) in the remaining epithelium, suggesting that a YAP/epiregulin/EGFR signaling regulates these cells. This is further supported by the Lats1/2 conditional mutants where aberrant nuclear Yap expression drives increased levels of epiregulin and ductal progenitors. The increase in progenitors is abrogated when EGFR signaling is blocked. Together these findings begin to define the role of Hippo pathway signaling in the development of the salivary gland as well as establishing the novel requirement for epiregulin.

Substantive concerns

1) Although this is the first study showing the effects of ablation or activation of the Hippo pathway on salivary gland development, the Hippo pathway has previously been examined in multiple epithelial organs and these have shown a requirement for this pathway in their development and progenitor cell populations. This study indicates that morphogenesis of the tissue as well as the progenitor populations are also perturbed in this organ but the manuscript falls short of revealing new insights into how YAP regulates morphogenic or stem cell programs. The authors do point out some very interesting phenotypic and cellular outcomes, such as the expanded ducts and polarity defects in the LATS1/2 conditional and reduced bud number in the YAP mutant, but stop short of investigating how these cell/tissue phenotypic outcomes are achieved. Further characterization of these phenotypes is required for publication.

2) The finding that epiregulin is regulated by Hippo signaling is novel. However, there is little data to support the claim that epiregulin is a regulator of ductal progenitors during gland development. The authors use a gain of function assay in the ex vivo system and show increased progenitor cells in the absence of YAP when treated with exogenous epiregulin. This indicates that epiregulin is sufficient, but it does not show that epiregulin is either necessary or is involved in this process in the wild type tissue. Unfortunately, use of a chemical inhibitor of EGFR in the LATS1/2 mutant cultured ex vivo does not solve this due to non-specificity. It is just as likely that inhibition of EGFR is perturbing HBEGF/EGFR interactions. Knockdown or ablation of epiregulin in the epithelium is needed.

3) Previous studies in the salivary gland show that parasympathetic nerves regulate KRT5+ progenitor cells (Knox et al., 2010; Knosp et al., 2015). The authors in the current study base their conclusion that parasympathetic nerves are not affected by ablation of YAP or are adversely affected by ablation of LATS1/2 by visual inspection. However, visual inspection at low resolution cannot be used alone to draw the conclusion that the nerves are or are not affected. qPCR analysis of neuronal proteins or neurotransmitters that are suggestive of function or growth is needed to substantiate the claim that nerves are or are not altered. In addition, ablation of the nerves or inhibiting of their function would also indicate that they are not involved in altering progenitor populations in the mutants.

Reviewer #3:

In this manuscript the authors report observations that advance our understanding of the role and regulation of Yap in the salivary gland. They show that Yap phosphorylation and localization are patterned, with nuclear (active) Yap in ductal progenitor cells, and cytoplasmic (inactive) Yap in differentiating cells. This is consistent with studies in other organs that have reported nuclear Yap in progenitor cells. They assess both loss and gain-of-function phenotypes for Yap, through deletion of a Yap conditional allele, or deletion of the Yap regulators Lats1 and Lats2. Consistent with the expression data, these studies support a key role for Yap in specifying progenitor cell fate, with both loss and gain of Yap activity severely disrupting salivary gland development. The authors also conduct expression studies, which leads them to identify the EGFR ligand Epiregulin as a key mediator of Yap activity in the salivary gland. An unanswered question is how Yap activity is normally controlled in the developing salivary gland, but I think it's reasonable to leave this for a future study. Overall I found this an interesting report that advances our understanding of salivary gland development and significantly advances our understanding of how Hippo signaling influences salivary gland development.

[Editors’ note: what now follows is the decision letter sent to the authors in response to their appeal.]

Thank you for choosing to send your work entitled "The Hippo pathway effector YAP is an essential regulator of submandibular gland ductal progenitor patterning" for consideration at eLife. Your letter of appeal has been considered by a Senior Editor and a Reviewing editor, and, based on their evaluation, was then sent to an additional expert reviewer for consideration. Having received and considered this fourth review, we are prepared to consider a revised submission with no guarantees of acceptance.

We request you meet the points discussed in your response to the reviewers’ comments as you propose; in particular, we note that genetic evidence supporting the proposed role of Epiregulin will be required for further consideration for publication in *eLife*. This point was also highlighted by the additional reviewer, who felt that SMG-specific Epiregulin knock out in the K14/K5 ductal progenitors and differentiated duct would significantly strengthen the manuscript. The additional reviewer also had a number of minor concerns, which we would like you to address in addition to the points discussed in your point-by-point response:

1. In the cartoon in Figure 1, please also highlight where cytoplasmic versus nuclear YAP is observed. In Figure 1, Krt5 is misspelled.

2. It is not clear what the marker is for the SMGs in Figure 2, left and middle panels, please clarify.

3. In Figure 6, if possible please show whether Ereg RNA is expressed in the same cells where Yap is nuclear, since there are good protocols for simultaneously imaging transcripts and protein.

4. Please provide a higher power image/zoom of the images in Figure 6, so that the reader can see if Yap is nuclear throughout the gland.

5. This reviewer also raised similar points to reviewer 2 regarding some of the images used, and also requested that details of the number of animals/samples assessed for each experiment should be included.

---

## [Author Response]

[Editors’ note: the author responses to the first round of peer review follow.]

*Reviewer #2:*

*[…] Substantive concerns*

*1) Although this is the first study showing the effects of ablation or activation of the Hippo pathway on salivary gland development, the Hippo pathway has previously been examined in multiple epithelial organs and these have shown a requirement for this pathway in their development and progenitor cell populations. This study indicates that morphogenesis of the tissue as well as the progenitor populations are also perturbed in this organ but the manuscript falls short of revealing new insights into how YAP regulates morphogenic or stem cell programs.*

We respectively disagree with this assessment from the reviewer. Our study reveals for the first time that Yap signaling is essential for salivary gland epithelial development, and offers novel insight into how Yap is directing the development of this organ. In doing so, we describe (for the first time) an important role for Epiregulin (Ereg) in SMG ductal progenitor patterning. Moreover, our manuscript describes (for the first time) that the Lats1/2 kinases are essential upstream regulators of Yap activity in SMG epithelium, and that loss of Lats1/2 leads to an uncontrolled expansion of Krt5/Krt14 cell populations. Our observations have implications beyond SMG development, as Krt5/Krt14 cell populations are critical for SMG epithelial repair and are often dysregulated in diseases, such as cancer. Therefore, we believe our work offers important novel insight into developmental and disease processes and will be relevant to a broad readership.

*The authors do point out some very interesting phenotypic and cellular outcomes, such as the expanded ducts and polarity defects in the LATS1/2 conditional and reduced bud number in the YAP mutant, but stop short of investigating how these cell/tissue phenotypic outcomes are achieved. Further characterization of these phenotypes is required for publication.*

Our manuscript describes Ereg as an important downstream target of Yap that is required for patterning of Krt5/Krt14 ductal epithelial cells, and notably we show that exposure of Yap-null SMGs to Ereg can partially rescue cell fate defects associated with these mutants. We agree that there is much more to be investigated based on the phenotypes we describe, but given that we characterize an important new mechanism for SMG development that contributes to one of the major defects observed in the Yap- and Lats1/2- knockout models, that being the ductal patterning defects, we argue that our manuscript offers important knowledge that should be published.

*2) The finding that epiregulin is regulated by Hippo signaling is novel. However, there is little data to support the claim that epiregulin is a regulator of ductal progenitors during gland development. The authors use a gain of function assay in the* ex vivo *system and show increased progenitor cells in the absence of YAP when treated with exogenous epiregulin. This indicates that epiregulin is sufficient, but it does not show that epiregulin is either necessary or is involved in this process in the wild type tissue. Unfortunately, use of a chemical inhibitor of EGFR in the LATS1/2 mutant cultured* ex vivo *does not solve this due to non-specificity. It is just as likely that inhibition of EGFR is perturbing HBEGF/EGFR interactions. Knockdown or ablation of epiregulin in the epithelium is needed.*

We have performed the Ereg knockdown experiment suggested by the reviewer, and have found that depletion of Ereg in ex vivo cultured SMGs leads to a dramatic loss of Krt5 and Krt14-positive cells and branching defects, agreeing with our model that Ereg is a key mediator of ductal progenitor specification. This data is shown in our new Figure 7.

*3) Previous studies in the salivary gland show that parasympathetic nerves regulate KRT5+ progenitor cells (Knox et al., 2010; Knosp et al., 2015). The authors in the current study base their conclusion that parasympathetic nerves are not affected by ablation of YAP or are adversely affected by ablation of LATS1/2 by visual inspection. However, visual inspection at low resolution cannot be used alone to draw the conclusion that the nerves are or are not affected. qPCR analysis of neuronal proteins or neurotransmitters that are suggestive of function or growth is needed to substantiate the claim that nerves are or are not altered. In addition, ablation of the nerves or inhibiting of their function would also indicate that they are not involved in altering progenitor populations in the mutants.*

We have revised how we describe the phenotypes we observe with respect to the parasympathetic nerve in our manuscript to better highlight that the nerves are innervated in Yap-null SMGs, but are not organized in the same pattern as in wild type SMGs. This observation is consistent with the defective patterning we observe in Yap-null these mutants, which we discuss in our manuscript.

[Editors’ note: the author responses to the re-review follow.]

*[…] We request you meet the points discussed in your response to the reviewer's comments as you propose; in particular, we note that genetic evidence supporting the proposed role of Epiregulin will be required for further consideration for publication in eLife. This point was also highlighted by the additional reviewer, who felt that SMG-specific Epiregulin knock out in the K14/K5 ductal progenitors and differentiated duct would significantly strengthen the manuscript. The additional reviewer also had a number of minor concerns, which we would like you to address in addition to the points discussed in your point-by-point response:*

*1. In the cartoon in Figure 1, please also highlight where cytoplasmic versus nuclear YAP is observed. In Figure 1, Krt5 is misspelled.*

As requested, we have updated our illustration in Figure 1 to highlight our observations with Yap localization. We also describe these observations in the updated figure legend, as well as in the main text.

*2. It is not clear what the marker is for the SMGs in Figure 2, left and middle panels, please clarify.*

We have updated Figure 2 to better describe the images shown.

*3. In Figure 6, if possible please show whether Ereg RNA is expressed in the same cells where Yap is nuclear, since there are good protocols for simultaneously imaging transcripts and protein.*

We have performed the experiment proposed by the reviewer, and found that cells expressing Ereg are cells with higher levels of nuclear Yap. We have included this data in our new Figure 6.

*4. Please provide a higher power image/zoom of the images in Figure 6, so that the reader can see if Yap is nuclear throughout the gland.*

As suggested by the reviewer we have updated the images in original Figure 6 with new zoomed in images that show predominantly cytoplasmic Yap in WT SMGs and nuclear Yap in *Lats1/2-cnull* SMGs. This data is included in our new Figure 8.

*5. This reviewer also raised similar points to reviewer 2 regarding some of the images used, and also requested that details of the number of animals/samples assessed for each experiment should be included.*

We have updated our representative figures along with the figure legends to better describe the numbers of repeats we performed for our experiments. We have also updated and added a description of the statistics we used in the respective figure legends.